



Geoscientific
Model Development

# APIFLAME v2.0 biomass burning emissions model: impact of refined input parameters on atmospheric concentration in Portugal in summer 2016

**Solène Turquety**[1], **Laurent Menut**[2], **Guillaume Siour**[3], **Sylvain Mailler**[2,4], **Juliette Hadji-Lazaro**[5], **Maya George**[5], **Cathy Clerbaux**[5,6], **Daniel Hurtmans**[6], and **Pierre-François Coheur**[6]

[1]LMD/IPSL, Sorbonne Université, ENS, PSL Université, École Polytechnique, Institut Polytechnique de Paris, CNRS, Paris, France
[2]LMD/IPSL, École Polytechnique, Institut Polytechnique de Paris, ENS, PSL Université, Sorbonne Université, CNRS, Palaiseau, France
[3]Laboratoire Interuniversitaire des Systèmes Atmosphériques (LISA), UMR7583, CNRS, Université Paris-Est-Créteil, Université de Paris, Institut Pierre Simon Laplace, Créteil, France
[4]Ecole des Ponts ParisTech, Université Paris-Est, 77455 Champs-sur-Marne, France
[5]LATMOS/IPSL, Sorbonne Université, UVSQ, CNRS, Paris, France
[6]Service de Chimie Quantique et Photophysique, Atmospheric Spectroscopy, Université Libre de Bruxelles (ULB), Brussels, Belgium

**Correspondence:** Solène Turquety (solene.turquety@lmd.polytechnique.fr)

**Abstract.** Biomass burning emissions are a major source of trace gases and aerosols. Wildfires being highly variable in time and space, calculating emissions requires a numerical tool able to estimate fluxes at the kilometer scale and with an hourly time step. Here, the APIFLAME model version 2.0 is presented. It is structured to be modular in terms of input databases and processing methods. The main evolution compared to version 1.0 is the possibility of merging burned area and fire radiative power (FRP) satellite observations to modulate the temporal variations of fire emissions and to integrate small fires that may not be detected in the burned area product. Accounting for possible missed detection due to small fire results in an increase in burned area ranging from $\sim 5\,\%$ in Africa and Australia to $\sim 30\,\%$ in North America on average over the 2013–2017 time period based on the Moderate-Resolution Imaging Spectroradiometer (MODIS) Collection 6 fire products.

An illustration for the case of southwestern Europe during the summer of 2016, marked by large wildfires in Portugal, is presented. Emissions calculated using different possible configurations of APIFLAME show a dispersion of 80 % on average over the domain during the largest wildfires (8–14 Au-

gust 2016), which can be considered as an estimate of uncertainty of emissions. The main sources of uncertainty studied, by order of importance, are the emission factors, the calculation of the burned area, and the vegetation attribution. The aerosol ($PM_{10}$) and carbon monoxide (CO) concentrations simulated with the CHIMERE regional chemistry transport model (CTM) are consistent with observations (good timing for the beginning and end of the events, $\pm 1$ d for the timing of the peak values) but tend to be overestimated compared to observations at surface stations. On the contrary, vertically integrated concentrations tend to be underestimated compared to satellite observations of total column CO by the Infrared Atmospheric Sounding Interferometer (IASI) instrument and aerosol optical depth (AOD) by MODIS. This underestimate is lower close to the fire region (5 %–40 % for AOD depending on the configuration and 8 %–18 % for total CO) but rapidly increases downwind. For all comparisons, better agreement is achieved when emissions are injected higher into the free troposphere using a vertical profile as estimated from observations of aerosol plume height by the Multi-angle Imaging SpectroRadiometer (MISR) satellite instrument (injection up to 4 km). Comparisons of aerosol layer

heights to observations by the Cloud-Aerosol Lidar with Or-
thogonal Polarization (CALIOP) show that some parts of the
plume may still be transported at too low an altitude. The
comparisons of the different CTM simulations to observa-
tions point to uncertainties not only on emissions (total mass
and daily variability) but also on the simulation of their trans-
port with the CTM and mixing with other sources. Consider-
ing the uncertainty of the emission injection profile and of the
modeling of the transport of these dense plumes, it is diffi-
cult to fully validate emissions through comparisons between
model simulations and atmospheric observations.

## 1   Introduction

Biomass burning is a major perturbation to atmospheric
chemistry, strongly contributing to the global budgets of
aerosols and trace gases. Emitted compounds significantly
alter air quality at regional scales (e.g., Heil and Golhammer,
2001; Keywood et al., 2015) and play a major role in the in-
terannual variability of the background atmospheric compo-
sition (e.g., Spracklen et al., 2007; Jaffe et al., 2008; Monks
et al., 2012). Through the emission of long-lived greenhouse
gases and aerosols and their interaction with radiation, they
also have an impact on climate. The availability of more and
more comprehensive databases of emission factors (e.g., An-
dreae and Merlet, 2001; Akagi et al., 2011) and of satellite
observations (e.g., Giglio et al., 2006) since the 1990s has al-
lowed for the development of emission inventories for a more
systematic integration of this large source in chemistry trans-
port models (CTMs). Here, version 2.0 of the APIFLAME
model (Turquety et al., 2014), developed for the calculation
of such an inventory for air quality applications, is presented.

Two main approaches have been developed to estimate
biomass burning emissions from satellite observations, based
either on the extent of the burned area (BA) or on the inten-
sity of the fire as estimated using the measured fire radia-
tive power (FRP). In both methods, emissions for a given
species are calculated as the product of the fuel consumed
(FC) and the emission factor (in gram species per kilogram
of dry matter burned) corresponding to the type of vegetation
burned. In the first approach, originally formulated by Seiler
and Crutzen (1980), the fuel consumed is obtained by multi-
plying the BA by the biomass density in the region affected
by fires, scaled by the fraction available for burning. Fuel
consumption is estimated based either on tabulated values
summarizing available experiments (e.g., Hoelzemann et al.,
2004; Mieville et al., 2010; Wiedinmyer et al., 2011) or on
simulations by carbon cycle and dynamic vegetation models
(e.g., van der Werf et al., 2010). More recently, "top-down"
approaches estimate the fuel consumed directly from FRP
observations, in particular to facilitate real-time applications
(Kaiser et al., 2012; Sofiev et al., 2009). The underlying hy-
pothesis is that the quantity of vegetation burned depends

on the intensity of the fire episode (Wooster et al., 2005).
In APIFLAME, the classical approach based on BA obser-
vations is used, but it was developed to allow for calcula-
tions from fire detection products available in near real-time.
It was constructed as a modular tool that can be adapted to
any user specification in terms of domain, horizontal reso-
lution, and chemical species. It was initially developed for
use with the CHIMERE CTM (Menut et al., 2013a; Mailler
et al., 2016), but it may easily be adapted to other model
specifications (chemical schemes) without modifications to
the source code. A full description of the model in its first
version is provided in Turquety et al. (2014) with an applica-
tion to fire emissions in Europe and the Mediterranean area.
It has been successfully used in different studies looking at
the impact of fires on the regional atmospheric composition
over Europe (Rea et al., 2015; Majdi et al., 2019), Australia
(Rea et al., 2016), California (Mallet et al., 2017), and Africa
(Menut et al., 2018).

The burned area processing method provided with the
code is based on the Moderate-Resolution Imaging Spec-
troradiometer (MODIS) fire observations of burned scars
(Giglio et al., 2018) and active fires (Giglio et al., 2006).
A major evolution since the first version of APIFLAME is
the possibility of merging both products in order to use the
day-to-day variability from the active fires (FRP dependent)
and/or include small fires that may not have been detected in
the burned scar product. Hourly variability based on geosta-
tionary observations (SEVIRI for Europe and Africa) is also
included. Emission factors have been updated according to
recently published data, and the possibility of using tabulated
fuel consumption from the literature has been added.

In this study, the ability of the APIFLAME model to pro-
vide useful information on emissions and the associated un-
certainty is analyzed for the case study of the 2016 fire sea-
son in southwestern Europe. It was marked by severe fires in
Portugal, where burned areas were twice the average over the
previous decade (while the number of fires remained stable)
(San-Miguel-Ayanz et al., 2017). Fire activity in other south-
ern countries was close to the average for previous years.
In order to simulate the influence of these large fires on at-
mospheric concentrations, the APIFLAME biomass burning
emissions are included in a simulation by the CHIMERE
CTM with meteorological simulations from the mesoscale
Weather Research and Forecasting (WRF) model.

The realism of the simulations is assessed by comparison
with available observations, focusing on aerosols and carbon
monoxide (CO) as the pollutants most impacted by fire emis-
sions. CTM simulations incorporating emissions are often
used to assess the quality of emissions, acting as an interme-
diary between emissions and atmospheric observations. This
approach is essential given the lack of in situ observations
close to the fires. It also has limitations since it will be in-
fluenced not only by emissions but also by the way these
are incorporated in the CTM, the simulated transport, and
the chemical evolution. For a fire event in Greece, Majdi

et al. (2019) showed that modifying the parameterization of emission injection heights or of secondary organic aerosol (SOA) formation mechanisms in a regional CTM results in a variation of up to 75 % for the surface $PM_{2.5}$ and 45 % for the aerosol optical depth (AOD). In the case of evaluations based on comparisons with remote sensing observations of aerosols, the calculation of optical properties adds to the total uncertainty (e.g., Majdi et al., 2020). Carter et al. (2020) analyzed the performance of different widely used inventories to simulate the impact of biomass burning on aerosol concentrations in North America using the GEOS-Chem model. They found a difference of up to a factor of 7 in total aerosol emissions from fires in North America mainly due to differences in the dry matter burned. The study also shows that performance within the CTM was dependent on the event and region considered, so it is not possible to identify a single inventory that provides the best agreement with observations for all cases. They also stress that some inventories were adjusted to allow better agreement between simulations and observations but that it may result in lower performance for other models or case studies (higher emissions can compensate for model biases due to other processes, for example).

The flexibility of the APIFLAME model allows an analysis of the sensitivity of the dry matter burned and the resulting emissions to the input databases used in the calculation. Here, different configurations of the emissions model are used to evaluate the sensitivity of the simulated concentrations to several key parameters: burned area, vegetation type, fuel consumption, and emission factors. In addition, the sensitivity to the parameterization of the fire emission injection heights is evaluated. This sensitivity analysis provides information on the uncertainty of the emissions and of the CTM simulations.

After a description of the general structure of the model (Sect. 2) and of the main input parameters (Sect. 3), the approach chosen for the merging of burned scars and active fires is described (Sect. 4). The application to the case of the 2016 Portuguese fires is then discussed (Sect. 5). The application of APIFLAME to that case study and the related uncertainty is first presented (Sect. 5.1). The available atmospheric observations of CO and PM are then presented (Sect. 5.2), as well as the CHIMERE CTM configuration used for the simulations (Sect. 5.3). The sensitivity of the simulated concentrations to the configuration of APIFLAME used and to injection heights is discussed in Sect. 5.4. The ability of the modeling system to provide information on regional air quality is then evaluated through comparisons with observations from surface networks (Sect. 5.5.1). Since few measurement sites are available, satellite observations of carbon monoxide (CO; IASI/MetOp-A,B), aerosol optical depth (AOD; MODIS/Terra), and aerosol layer height (MISR/Terra and CALIOP) (Sect. 5.5.2) are used for a regional analysis of the simulated daily variability and the fire plume transport (Sect. 5.5.3).

## 2 Model general structure

The general principle of the calculation is sketched in Fig. 1. The code was designed to be modular, allowing different user choices for sensitivity analyses. For each fire detected, the corresponding emissions $E_i$ (g) for a chemical species $i$ is calculated as follows:

$$E_i = A \sum_{v=1}^{\text{veg types}} f_v F_v \epsilon_{v,i}, \qquad (1)$$

where $A$ (m$^2$) is the burned area, $f_v$ is the fraction of this surface in vegetation type $v$, $F_v$ is the biomass consumed (kilograms of dry matter, DM; kg m$^{-2}$) for this vegetation type, and $\epsilon_{v,i}$ (g kg DM$^{-1}$) is the emission factor corresponding to species $i$ and vegetation $v$. Any species may be added to the inventory provided its emission factor is known. The vegetation type is attributed fire by fire before being gridded onto the specified grid (domain and associated horizontal resolution). This allows high-resolution calculations that will keep the variability from the fire and vegetation datasets.

In addition to the hourly emissions for the model species selected (cf. Sect. 3.4), the grid cell area and the FRP, which gives an indication of fire intensity, are also provided. Coincident FRP values may be useful for plume height modeling. For each grid cell, the maximum FRP is calculated, as well as the statistical distribution of specified FRP bins. However, burned area and FRP are not always detected at the same time or location in the MODIS datasets. If the burned area dataset (MCD64 product; cf. Sect. 3.1) is chosen, there may be grid cells with nonzero burned area (nonzero emissions) but zero FRP (or the other way around). Merging both datasets may be an interesting option for some applications, for example, to improve temporal variability or in order to avoid missing small fires that may not have a detectable signature on both products (cf. Sect. 4).

The general structure has slightly changed compared to APIFLAME v1.0: the gridding onto the chosen model domain is now performed on the burned area before the calculation of emissions. This does not change results since the vegetation fraction is also gridded at this stage. However, the subsequent calculation of emissions is much faster once this initial step is done, allowing fast calculation of an ensemble of emissions using different configurations of APIFLAME. This provides valuable information on the possible uncertainty of the emissions.

## 3 Input observations and databases

The datasets required to compute fire emissions are briefly described below. Compared to APIFLAME v1.0, the code has been updated to use the MODIS Collection 6 data, the emission factors table has been updated, and the possibility of using fuel consumption from the literature has been added.

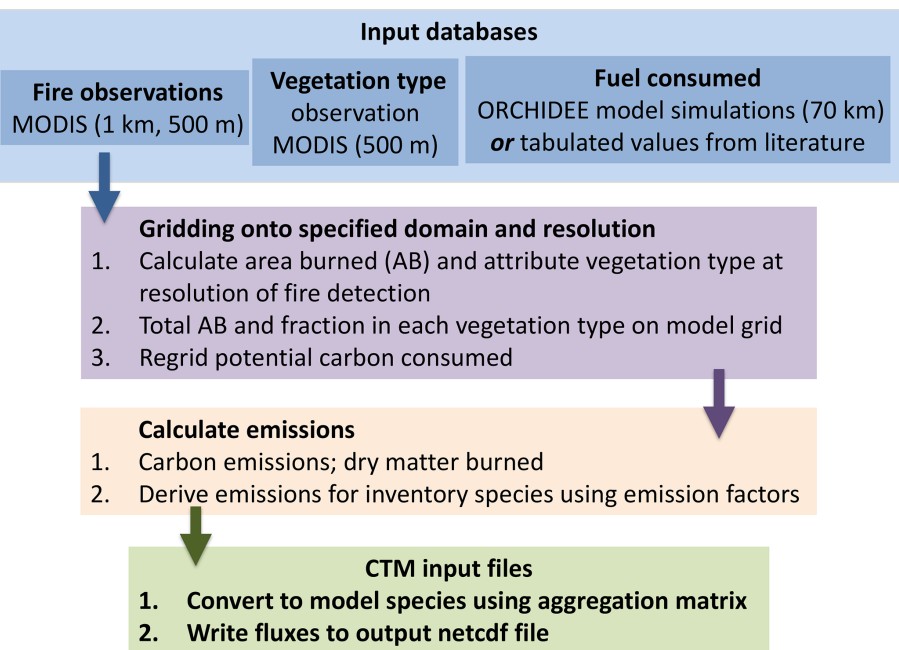

**Figure 1.** Overview of the APIFLAME v2.0 emissions model.

### 3.1 Fire observations

Although adaptable to any burned area database, API-FLAME was developed to derive BA from the MODIS fire databases (Collection 6; https://lpdaac.usgs.gov/dataset_discovery/modis/modis_products_table/, last access: 12 February 2020). The MCD64A1 burned scars product, based on the alteration of the surface reflectance (Giglio et al., 2010, 2015, 2018), provides the date of burning at 500 m horizontal resolution. The active fire products, MOD14 for MODIS/Terra (Equator overpass time 09:30 and 22:30 LT) and MYD14 for MODIS/Aqua (Equator overpass time 13:30 and 01:30 LT), based on thermal anomalies, are also used. These products provide the FRP at 1 km resolution (Giglio et al., 2006).

Only high confidence active fire detections are considered (quality index > 8), but false detections mainly associated with industrial activity may remain. These are filtered as described in Turquety et al. (2014). A fire pixel is rejected if the corresponding vegetation type is more that 50 % urban (fraction may be modified depending on situations), if it is located less than 1 km from an active volcano, or if the frequency of burning is unrealistically high at this location ($\geq 40$ %) based on the climatology of MODIS active fire detection. On the other hand, burned scars may miss smaller fires which are more easily detected by their thermal signature (Randerson et al., 2012).

Both MODIS datasets are systematically processed to derive the burned area, either using one dataset alone or merging both datasets as described in Sect. 4, and to allow users to use the FRP for other possible applications in their analysis.

For example, it is used as information on fire intensity for the calculation of plume injection heights by pyroconvection in several schemes (e.g., Sofiev et al., 2012).

In order to access information on the diurnal variability, SEVIRI and MSG data from the geostationary Meteosat Second Generation (MSG) satellite may be used for Europe and Africa (http://landsaf.ipma.pt/en/products/fire-products/frppixel/, last access: 24 June 2020, full MSG disk database). The active fire products also include the FRP, provided at 15 min temporal resolution, for pixels of $\sim 3$ km horizontal resolution at nadir (Roberts et al., 2005).

For all products, uncertainty is mainly due to cloud cover, which prevents the observation of surface anomalies. The uncertainty of the temporal variability derived from the MCD64A1 burned area is estimated at about 2 d based on coincidences with active fires (Giglio et al., 2018). The high temporal coverage of the SEVIRI observation increases the probability of detecting a fire, but the larger pixel size also increases the limit of detection so that small fires may be missed.

### 3.2 Vegetation cover

For the calculation of the burned area from fire detection, the MODIS vegetation cover fraction (VCF) product (MOD44B v006) is used. It provides the fraction of tree and non-tree vegetation cover for 250 m × 250 m pixels, which is converted to 500 m and 1 km resolutions for compatibility with the fire products. Only the fraction of MODIS fire pixels covered by vegetation is assumed to burn.

The vegetation type burned is also important to derive the fuel consumed and attribute the emission factors for each emitted species. Three land cover datasets may be used: the USGS and the CORINE Land Cover (CLC) datasets, constructed at 1 km resolution (cf. Turquety et al., 2014, for details), and the MODIS vegetation classification. The MODIS land cover type product (MCD12Q1 v006) provides information on the land cover at 500 m resolution, specific to the year analyzed, which is associated with each burning pixel during the burned area processing. Both MODIS vegetation products may be retrieved from the NASA LPDAAC (https://lpdaac.usgs.gov/dataset_discovery/modis/modis_products_table, last access: 3 February 2020). The vegetation types attributed to the burned area on the user-specified grid are provided in the model output files.

### 3.3 Biomass density and fuel consumed

Biomass density available for burning is derived from simulations by the ORCHIDEE model (Maignan et al., 2011). The fraction susceptible to burning is calculated based on tabulated fractions for each plant functional type (PFT) and carbon pool (litter, wood, leaves, and roots), scaled according to plant moisture stress. The method chosen is described in detail in Turquety et al. (2014). Monthly averaged fields are interpolated to the user-defined grid at the beginning of the simulation. The current APIFLAME archive provides a monthly averaged fuel consumption climatology constructed from global ORCHIDEE simulations for the period 1989–2008 at 70 km resolution.

The possibility of using tabulated values is also implemented in version 2 of the code. Fuel consumption from van Leeuwen et al. (2014), compiled from measurements published in peer-reviewed literature, is then used by default. Table 1 reports the values calculated with APIFLAME and the tabulated values for different biomes. A good agreement in the average values is obtained for all biomes except tropical forests, for which fuel consumption is strongly underestimated. Here no wood is considered to be burning for forest types in the fuel-consumed calculation, although it represents a large fraction of carbon density. The contribution from this carbon pool might be underestimated for tropical forests. For regions or case studies strongly affected by tropical forest fires, users are advised to use the tabulated values. Elsewhere, the APIFLAME approach based on ORCHIDEE simulations is preferable since it allows for more variability in space and time (monthly). For peatlands, values from the literature are used by default.

### 3.4 Emission factors and emitted species

Emission factors from Akagi et al. (2011) are used, including updates from Yokelson et al. (available at http://bai.acom.ucar.edu/Data/fire/, last access: 24 June 2020) (Yokelson et al., 2013; Akagi et al., 2013; Stockwell et al., 2014, 2015).

Although emission factors strongly depend on the phase of combustion (flaming favoring $CO_2$, $NO_x$, and $SO_2$; smoldering favoring CO, $CH_4$, $NH_3$, non-methane volatile organic compounds (NMVOCs), and organic aerosols, for example), the reported emission factors (EFs) used in the model correspond to the average amounts for different fire-type categories. Since the inventory aims at being used in air quality models at resolutions of several kilometers, both flaming and smoldering phases should be mixed in one grid cell so that using average values is relevant.

The emission factors included in this version are provided in Tables S1 and S2 in the Supplement. In the code, these are provided in a dedicated input file that may easily be modified by users according to specific needs. Several families grouping volatile organic compounds (VOCs) are considered:

- alkan – butane and higher alkanes (molar weight MW = $58\,\mathrm{g\,mol^{-1}}$);

- alken – butene and higher alkenes (MW=$56\,\mathrm{g\,mol^{-1}}$);

- other alcohols – all non-$CH_3OH$ alcohols: ethanol and higher (MW = $46\,\mathrm{g\,mol^{-1}}$);

- other aldehydes – all non-$CH_2O$ aldehydes: $CH_3CHO$ and higher (MW = $44\,\mathrm{g\,mol^{-1}}$);

- other ketones – all non-acetone ketones (MW = $72\,\mathrm{g\,mol^{-1}}$);

- arom – other aromatics (MW = $126\,\mathrm{g\,mol^{-1}}$);

- furans – all furans (MW = $82\,\mathrm{g\,mol^{-1}}$).

In order to convert emissions from inventory species (for which an emission factor is provided) to model species (needed for model simulations, depending on the chemical scheme), aggregation matrices are used. For VOCs, the emissions for listed compounds are lumped into a smaller set of model compounds using a reactivity weighting factor accounting for the relative rate constants for reactions with the OH radical following Middleton et al. (1990). Aggregation matrices are provided for the Melchior (Derognat et al., 2003) and SAPRC-07-A (Carter, 2010) mechanisms used in the CHIMERE model. If another scheme is considered, a new aggregation matrix should be constructed (input files independent from the core of the model).

For aerosol species, a surrogate species ("other PPM", primary particulate matter) is introduced to fill the gap between the sum of primary emitted species identified and the reported amounts of $PM_{2.5}$ (note that $1.6 \times E_{OC}$ is removed as organic carbon (OC) is increased by 60 % in the aggregation step in order to account for fast chemistry). Majdi et al. (2019) show that this additional mass could correspond to secondary aerosol (SOA) formation from intermediate and semi-volatile organic compounds that are usually not well accounted for in CTMs. If users need/want to add new inventory species, the relevant emission factors have to be added

**Table 1.** Fuel consumption (in kilograms dry matter per square meter burned) calculated in APIFLAME from ORCHIDEE simulations (on average for fires detected in 2013–2017) and values reported in van Leeuwen et al. (2014) (used in tabulated approach). The standard deviation is provided in parentheses.

| Biome | APIFLAME/ORCHIDEE | Values from van Leeuwen et al. (2014) |
|---|---|---|
| Tropical forest | 2.6 (1.7) | 12.6 (7.7) |
| Temperate forest | 3.3 (1.8) | 5.8 (7.2) |
| Boreal forest | 4.7 (2.0) | 3.5 (2.4) |
| Savanna | 0.3 (0.3) | 0.46 (0.22) |
| Grassland savanna | 0.9 (1.0) | 0.43 (0.22) |
| Wooded savanna | 1.9 (1.3) | 0.51 (0.22) |
| Pasture | 1.0 (1.0) | 2.8 (0.93) |
| Cropland | 1.0 (1.1) | Shifting cultivation: 2.3 (–) |
|  |  | Crop residue: 0.65 (0.9) |
| Chaparral | 1.6 (1.5) Shrubland, wooded savanna at midlatitudes | 2.7 (1.9) |
| Tropical peatland | – | 31.4 (19.6) |
| Boreal peatland | – | 4.2 (–) |
| Tundra | 3.1 (2.3) Shrubland, savanna, grassland at latitudes > 50° N | 4.0 (–) |

to the emission factor list, and the aggregation files need to be updated.

## 3.5 Correspondence between vegetation types, ecozones, and PFTs

Calculation of the emissions requires information on the type of vegetation burned. It is attributed using a given land cover database. The present version of the code allows for the use of two databases at a global scale (MODIS and USGS) and an additional one for Europe (CLC), described in Sect. 3.2.

If a regional database is chosen, it may be complemented by one of the global databases. In the code, when CLC is chosen for Europe, MODIS is taken by default for regions not covered by the CLC database. For this purpose, a matrix of correspondence between the MODIS International Geosphere–Biosphere Program (IGBP) and the CLC vegetation types is provided. When there is no direct correspondence between the two land cover datasets, their description was used. For example, for IGBP vegetation type "woody savanna", described as 30 %–60 % tree cover, the CLC correspondence chosen is 30 % mixed forest and 70 % natural grassland.

The vegetation type is attributed during the burned area preprocessing. Correspondence matrices between vegetation types and ORCHIDEE plant function types (PFTs) and between vegetation types and the ecozone in the emission factor listing are used to allow consistent calculations. These matrices are quite subjective and may be modified for tests or depending on the region considered. Also, if a different database is used for vegetation attribution in the BA processing, new matrices have to be constructed.

Depending on the vegetation database, vegetation in regions with chaparral, bushes, or Mediterranean vegetation types (Mediterranean area, California, Australia) may be classified as shrubland, wooded savanna, or savanna. For these regions and vegetation types, and to limit inconsistencies in APIFLAME, the fire type is classified as chaparral, and the fuel load is calculated using both forest and grass PFTs in the corresponding grid cells.

## 4 Daily and hourly temporal variability of burned area

### 4.1 Merging burned scars and active fire products

The fire observations described in Sect. 3.1 provide a date of burning at a resolution of 500 m (MCD64 burned scars product) or 1 km (MOD14 active fire product). The burned area is calculated for each burning pixel in the database as the pixel area actually covered by vegetation (cf. Sect. 3.2), as in the first version of APIFLAME (Turquety et al., 2014).

Differences in the location of burned areas and active fires detected were found for different events in different regions. While the APIFLAME methodology is based on the burned area, a combination of the estimated burned area with the FRP product is also proposed in version 2. This option offers the possibility of relying primarily on the total monthly burned area from the MCD64 product (burned scar) but to redistribute it temporally depending on the fire intensity. While the total burned area (and thus the total emissions) will remain the same, the emissions will peak when FRP is largest. For each grid cell $i$, the burned area during day $d$ is the following:

$$A_{\mathrm{merged}}(i,d) = \frac{\mathrm{FRP}(i,d)}{\sum_{t=0}^{\mathrm{nd}}\mathrm{FRP}(i,d)} \sum_{t=0}^{\mathrm{nd}} A(i,d), \qquad (2)$$

with "nd" being the number of days in the current month.

If this option is chosen, there will be no modification of the daily variability of the BA in grid cells with no coincident active fire. There may also be grid cells with significant

FRP values but no burned scar detected. This will, in particular, be the case for small fires (Randerson et al., 2012). The approach chosen in Randerson et al. (2012) derives a small fire BA in each region using an average burned area per active fire, which was calculated for each region based on coincident MCD64 and MOD14 detections. It is then scaled according to the amplitude of the variations of surface reflectance (providing information on fire intensity). To limit the number of datasets required to run APIFLAME, a simple linear modulation based on the FRP is used. Small fires are only included if the maximum FRP in the corresponding grid cell is > 50 MW, and the full pixel ($\sim 10^6 \, \mathrm{m}^2$) is allowed to burn only if the maximum FRP is > 1000 MW (which corresponds to extreme values, $\sim$ 99th percentile of the global FRP dataset for 2013–2017) so that the burned area from small fires in grid cell $i$ during day $d$ is estimated from the number of active fire detections that are not collocated with burned scars, $N_{out}(i, d)$, as follows:

$$A_{small}(i, d) = N_{out}(i, d) \times 10^6 \times \frac{\mathrm{FRP}_{max}(i, d)}{\Delta \mathrm{FRP}}, \qquad (3)$$

with $\Delta \mathrm{FRP} = 1000$ MW. A larger burned area is therefore associated with fires of greater radiative intensity. This may not be true if the satellite overpass coincides with the flame phase of the fire. Even a small fire can then have a high FRP. However, intense fires are expected to burn more fuel. The reported burned area should therefore be analyzed either as a larger area or as a larger burning fraction. This follows the same logic as the merging of burned scars and active fires.

The choice in final burned area is left to the user: burned scar (MCD64), active fires (MOD14), merged burned scar and active fires according to Eq. (2), and merged product including small fires. These options may be used to analyze the possible uncertainty of the emissions.

Figure 2 shows the spatial distribution of the yearly average fire activity for 2013–2017. The monthly variability over different regions is provided in the Supplement (Fig. S1).

The regional and temporal variability for this specific time period is consistent with previous analyses (e.g., Giglio et al., 2010; Earl and Simmonds, 2018) with large and frequent burning in tropical regions and more sporadic events in temperate and boreal regions. Fire seasons coincide with the dry seasons in most regions: maximum in winter in the northern tropics, in August–October in the southern tropics, and during boreal summer in the middle and high latitudes of the Northern Hemisphere. The interannual variability is high in most regions except in Africa and South America due to lower variability in rainfall and the use of burning for land management (slash and burn agriculture). The El Niño–Southern Oscillation (ENSO) explains a large part of the observed variability. El Niño years (weak in 2014–2015, very strong in 2015–2016) result in particularly dry conditions in southeast Asia but also in Australia and Alaska, resulting in more severe burning seasons (e.g., Earl and Simmonds, 2018).

The modulation using the FRP value used in this study is a strong approximation that should be used with caution. It results in a large potential increase in calculated burned area over most regions: $\sim$ 46 % in boreal and in temperate North America, 37 % in equatorial Asia, and 16 %–22 % in boreal Asia, southeast Asia, Europe, and Central and South America. In Africa and Australia, calculated contributions are lower ($\sim$ 5 %) due to the low FRP of active fires not collocated with an MCD64 detection. Randerson et al. (2012) estimate an increase in burned area from small fires ranging from 7 % in Australia (5 % in this study) to 157 % in equatorial Asia (37 % in this study) for the 2001–2010 time period based on the MODIS Collection 5 fire products. They generally found much higher contributions, for example, in temperate North America (75 %) and Europe (112 %) and in boreal Asia (62 %). However, the Collection 6 product (used here) has been shown to detect more fires (26 % increase in global burned area over the period 2001–2016) with better coincidence with active fire products (68 % within 2 d) (Giglio et al., 2018). For example, mean annual burned area for the period 2002–2016 in Europe is 71 % higher in Collection 6 than in Collection 5.

Since the methodology used here relies on the burned area (Eq. 1), the increase in burned area by small fires directly affects emissions. However, the relationship is not linear as it depends on the vegetation type. For CO, for example, emissions are increasing by $\sim$ 60 % in North America, $\sim$ 20 % in Central and South America, $\sim$ 25 % in southeast Asia, $\sim$ 23 % in Europe, and $\sim$ 5 % in Africa and Australia.

## 4.2 Hourly variability

Information of the diurnal variability of emissions has been shown to be critical, in particular to simulate the impact on regional air quality (e.g., Rea et al., 2016). This information is provided by instruments carried on board geostationary platforms (active fire observations, including FRP). In APIFLAME, users may chose to use either no diurnal variability (emissions constant during the day), an averaged hourly profile (Turquety et al., 2014), or an hourly profile derived from the scaled diurnal variability of FRP ($\mathrm{FRP}_{geo}$). Once the total daily emissions are calculated in a given grid cell $i$ for a given day $d$, the fraction $f_{hourly}$ emitted at hour of day $h$ is

$$f_{hourly}(i, d, h) = \frac{\mathrm{FRP}_{geo}(i, d, h)}{\sum_{h=1}^{24} \mathrm{FRP}_{geo}(i, d, h)}. \qquad (4)$$

One difficulty is that the horizontal resolution of geostationary observations is coarser ($\sim$ 3 km), and thus the probability of having a cloudy pixel is higher in spite of the good temporal revisit. Therefore, active fire observations from instruments on polar orbiting platforms, like MODIS, and geostationary platforms may not agree in location. The approach we have chosen is to use the spatial and daily variability from the MODIS product and apply a regional diurnal profile, calculated based on geostationary observations at coarser hori-

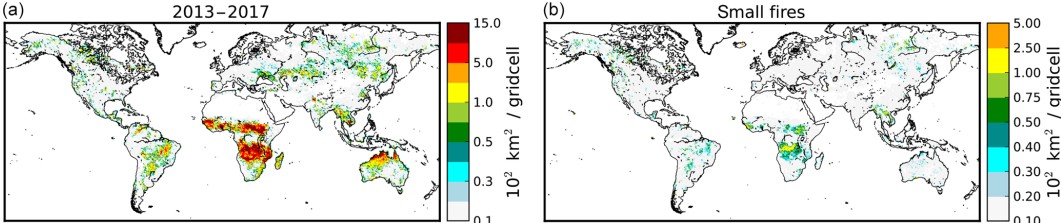

**Figure 2.** Average yearly burned area during the period 2013–2017 within $0.5° \times 0.5°$ grid cells as derived from the MODIS burned area product MCD64 **(a)** and including small fire contributions from MODIS active fire detection (MOD14) only **(b)**.

zontal resolutions to maximize the probability of having co-incidences. Two options are coded in APIFLAME: use SE-VIRI hourly variability at the same resolution as the resolution chosen for the final emissions or use a fixed $1° \times 1°$ res-olution. The second method implies that the same, averaged, diurnal profile will be used for $1°$ resolution regions even if a smaller horizontal resolution is chosen for the calculation of emissions. If no coincidence between MODIS and SEVIRI fires is obtained, no diurnal variability is applied (constant emissions during the day).

To further smooth possibly artificially high hourly vari-ability, daily data are processed in three steps:

1. The data are gridded at hourly resolutions on the model grid and a fixed $1° \times 1°$ grid.

2. Gaps shorter than 5 h between two detections of more than 1 h are filled using linear interpolation.

3. The resulting hourly distribution is smoothed using a polynomial fit.

Examples are shown in Fig. 3 for large fires in Portugal dur-ing the summer of 2016. Even after an averaging of the data on larger grid cells, the temporal variability at 15 min tem-poral resolution seems unrealistic. Averaging at 1 h temporal resolution allows a first smoothing of the dataset, but there are still gaps ($FRP_{geo} = 0$) between two periods of fire ac-tivity. The linear interpolation fills in smaller gaps ($\leq 4$ h, as seen in panel a), while the smoothing fills larger gaps but can strongly decrease the peak values (panel b).

## 5 Application to the summer of 2016 in southwestern Europe

The use of APIFLAME emissions in different configurations within a CTM allows for an evaluation of the impact of fires on atmospheric chemistry and of the associated uncertainty. This part describes an application to fires in Portugal dur-ing the summer of 2016. After a description of the calcu-lated emissions for this event, the atmospheric observations of trace gases and aerosols used for evaluation, as well as the model simulations performed, are described. The analysis of the simulations is focused on the increase due to the wildfire

event. A sensitivity study allows an analysis of the influence of several key factors in the calculation of fire emissions: burned area, vegetation type, emission factors, and emission injection profiles. The simulations are then evaluated through comparisons to surface and satellite observations.

### 5.1 APIFLAME biomass burning emissions

The MODIS observations of burned area and maximum FRP during June–September 2016 are mapped in Fig. 4. The largest fires affected the northern and central regions of Por-tugal, with 92 % of the total burned area according to the for-est fire report for the 2016 fire season in Europe by the Euro-pean Forest Fire Information System (EFFIS) (San-Miguel-Ayanz et al., 2017). More than 70 % of the total burned area that summer occurred in August. The daily burned area ob-tained with APIFLAME using different processing configu-rations during August 2016 is shown in Fig. 5.

In Portugal, the total burned area is 99 849 ha using the MCD64 product and 108 962 ha for the MERGE approach (144 882 ha including small fires). The difference between the MERGE and MCD64 totals is due to the fact that the burned area associated with small fires was included if there were active fires in a grid cell during the considered time period but no MCD64 burned area. The EFFIS report in-dicates a total of 115 788 ha burned during August 2016 (San-Miguel-Ayanz et al., 2017). This suggests that includ-ing small fires for this region results in an overestimate of the burned area.

The vegetation type burned can be attributed using either the MODIS IGBP or the CLC classification. For Portugal with the MODIS IGBP classification, 15 % of the MCD64 burned area is attributed to forest, 47 % to wooded savanna, and 36 % to savanna and grassland. Using the CLC land cover, 83 % is attributed to forest and 13 % to artificial. About the same distribution holds for small fires. According to the EFFIS report for the year 2016, 52 % burned in wooded land, mostly eucalyptus and pine stands, and 48 % in shrubland. In the IGBP classification, shrubland corresponds to woody vegetation with height $< 2$ m, while savanna corresponds to herbaceous or other understory vegetation with forest cover $< 30$ % (10 %–30 % for woody savanna) with height $> 2$ m. The different definitions of classes explain the different types

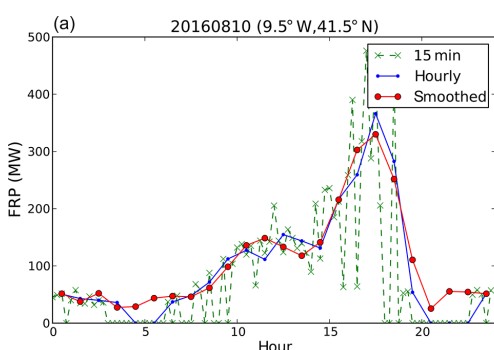
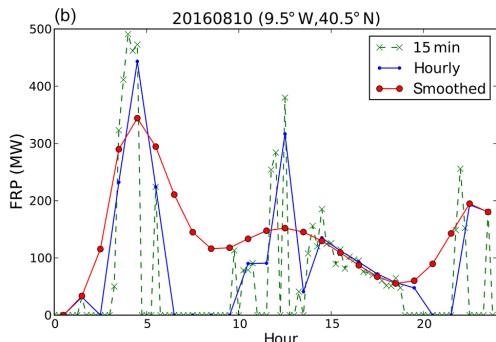

**Figure 3.** Example of SEVIRI FRP observations during wildfires in northern Portugal in 2016 (case study discussed in the following), averaged in $1° \times 1°$ grid cells at 15 min temporal resolution and 1 h temporal resolution and smoothed using a gap filling procedure followed by a polynomial fit.

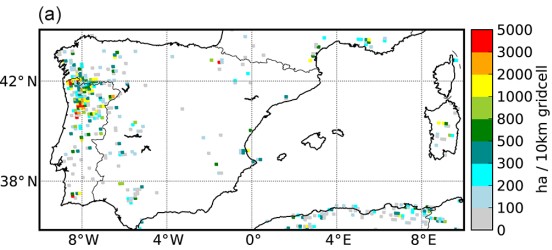
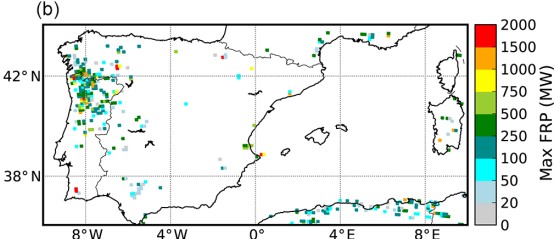

**Figure 4.** Total burned area derived from the MODIS MCD64A1 product and mapped on a 10 km resolution grid **(a)** and maximum FRP from the MODIS MOD14 product on the same grid **(b)** for observations from June to September 2016.

of vegetation burned here. However, it adds difficulty in the calculation of the resulting emissions.

The EF list in APIFLAME may be modified to include values reported for specific regions. Here, we include an experiment using the values reported by Alves et al. (2011b) for evergreen forest fires in Portugal in May 2009. These are used for both temperate forest and chaparral. Alves et al. (2011b) report emission factors of $170 \pm 83 \, \mathrm{g\,kg^{-1}}$ dry matter (DM) for CO (almost twice as large as those used here for temperate forests), of $14 \pm 4.5 \, \mathrm{g\,kg^{-1}}$ for $PM_{10}$ (slightly lower than the value of $17.7 \, \mathrm{g\,kg^{-1}}$ used here for temperate forest), and of $12 \pm 3.3 \, \mathrm{g\,kg^{-1}}$ for $PM_{2.5}$ (in agreement with the value of $12.8 \, \mathrm{g\,kg^{-1}}$ used here). Reisen et al. (2018) report emission factors of $PM_{2.5}$ for prescribed burns in eucalypt forests in southern Australia of $16.9 \, \mathrm{g\,kg^{-1}}$ DM during flaming combustion and $38.8 \, \mathrm{g\,kg^{-1}}$ DM during smoldering combustion. The recent inventory by Andreae (2019) reports an average of $113 \pm 50 \, \mathrm{g\,kg^{-1}}$ DM for CO in temperate forests and $18 \pm 14 \, \mathrm{g\,kg^{-1}}$ DM for $PM_{2.5}$. The values of Alves et al. (2011b) used here for the sensitivity simulation are on the higher end of estimates for CO but quite conservative for aerosols.

In order to test the sensitivity of the model to the main factors, biomass burning emissions for Portugal have been calculated using different configurations of APIFLAME. The tests are summarized in Table 2, and the corresponding impact on total CO and organic carbon (OCAR species in CHIMERE) emissions are reported in Table 3. The daily variation in CO biomass burning emissions is presented in Fig. 6.

Emissions using the same burned area processing configurations but different vegetation databases can show significant differences in magnitude and temporal variations. Merging burned area and FRP (BA-FRP) compared to using the burned area data alone results in an increase of 8 %–10 % of emissions. Adding the contribution from small fires to the MCD64 burned area (BA-sf) results in an increase of 33 %–36 %.

Using the tabulated fuel consumption has a low impact ($\pm 2$ %). This option is only available with the MODIS IGBP vegetation type. If the vegetation type is forced to temperate forest, the vegetation type burned according to CLC, the resulting emissions are close to the BA-sf-CLC configuration. This demonstrates the good consistency of the fuel consumption calculation in APIFLAME.

Emissions based on the CLC vegetation database are about 17 % higher than emissions based on the MODIS vegetation for CO. For OCAR, using CLC results in a small decrease in the estimated emissions. For other PPM, emissions are much lower using CLC due to lower emission factors for temperate forest than for pasture maintenance, chaparral, and savanna.

The strongest impact during this event corresponds to the choice of emission factor database. Using the values reported for forest fires in Portugal increases total CO emissions by 126 % for CO and 50 % for OCAR.

https://doi.org/10.5194/gmd-13-1-2020 Geosci. Model Dev., 13, 1–29, 2020

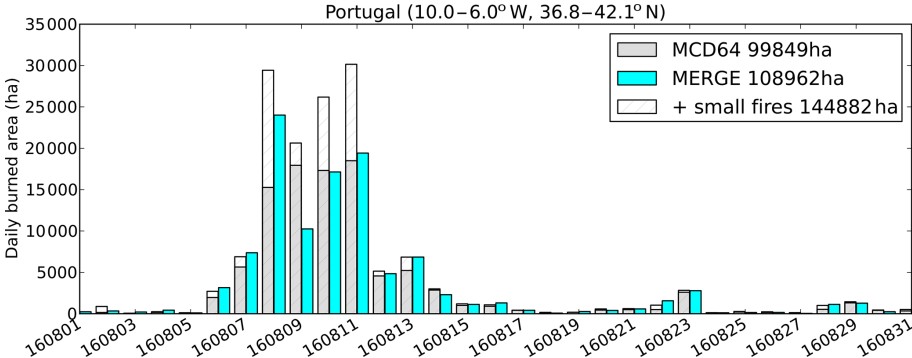

**Figure 5.** Total daily burned area during July–August 2016 over northern Portugal. The three different configurations available in APIFLAME are shown: MCD64A1 product alone (BA), MDC64A1 monthly total with daily variability depending on the MOD14 active fire product (MERGE; BA-FRP), and the MCD64A1 product including small fires (+ small fire; BA-sf).

**Table 2.** Scenarios used in APIFLAME for the calculation of emissions during the summer of 2016.

| Name | Burned area[a] | Vegetation type | Fuel consumed |
|------|------------|-----------------|---------------|
| BA-CLC | MCD64 | CORINE Land Cover (CLC) | ORCHIDEE |
| BA-MODIS | MCD64 | MODIS | ORCHIDEE |
| BA-FRP-CLC | Merge MCD64 with MOD14 FRP | CLC | ORCHIDEE |
| BA-FRP-MODIS | Merge MCD64 with MOD14 FRP | MODIS | ORCHIDEE |
| BA-sf-CLC | MCD64 + small fires | CLC | ORCHIDEE |
| BA-sf-MODIS | MCD64 + small fires | MODIS | ORCHIDEE |
| BA-sf-MODIS-lit | MCD64 + small fires | MODIS | van Leeuwen et al. (2014); cf. Table 1 |
| BA-sf-MODIS-lit-forest | MCD64 + small fires | MODIS | van Leeuwen et al. (2014); cf. Table 1 for temperate forest |
| BA-FRP-MODIS-EF | Merge MCD64 with MOD14 FRP | MODIS | Alves et al. (2011b) for forest; cf. text for details; emission factors CO, OC, BC |
| BA-sf-MODIS-EF | MCD64 + small fires | MODIS | Alves et al. (2011b) for forest; cf. text for details; CE1 emission factors CO, OC, BC |

[a] Calculated using MODIS burned scar (MCD64) or active fire (MOD14) products.

The dispersion of the regional daily total emissions is quantified as the average coefficient of variation (CV = standard deviation / mean value). This provides information on the uncertainty of emissions. Considering all experiments, the CV on the total daily emissions of CO, averaged over the duration of the fire event, is equal to 40 %. Without the experiments including small fires, it is 15 %. Without the experiments with emission factors from Alves et al. (2011b), it is 20 %. At 10 km resolution over the full domain (hence without summing all emissions within the region), the average CV is around 80 % of daily emissions, 60 % without experiments including small fires, and 76 % without experiments on emission factors.

## 5.2 Observations of atmospheric concentrations

Measurements of CO, $PM_{10}$, and $PM_{2.5}$ from the European air quality database (AirBase, https://www.eea.europa.eu/data-and-maps/data/airbase-the-european-air-quality-database-7, last access: 13 July 2018) are used for the validation of simulated surface concentrations. Only rural or suburban background

**Table 3.** Relative impact (%) of factors tested in the calculation of the emissions with APIFLAME on the total emissions of CO and organic carbon and on the resulting concentrations. Differences of simulated concentrations ($\Delta$CO and $\Delta$PM$_{10}$) are calculated as $(X_{test} - X_{ref})/X_{ref}$ and then averaged over points with relative impact from fires $> 10\%$.

| Factor | Simulations compared | $\Delta E_{CO}$ | $\Delta E_{OCAR}$ | $\overline{\Delta CO}$ | | $\overline{\Delta PM_{10}}$ | |
|---|---|---|---|---|---|---|---|
| | | | | surface | total | surface | total |
| Merge BA-FRP | BA-FRP-MODIS − BA-MODIS | +10 | +8 | – | – | – | – |
| Small fires | BA-sf-MODIS − BA-MODIS | +33 | +36 | +43 | +48 | +41 | +47 |
| Vegetation | BA-sf-CLC − BA-sf-MODIS | +17 | −0.3 | −30 | −17 | −11 | −29 |
| Fuel consumption | BA-sf-MODIS − BA-sf-MODIS-lit | +2 | −2 | – | – | – | – |
| Emission factor | BA-sf-MODIS-EF − BA-sf-MODIS | +126 | +50 | +152 | +118 | +40 | +14 |
| Injection height | BA-sf-MODIS MISR − BA-sf-MODIS | – | – | −25 | +11 | −22 | +32 |

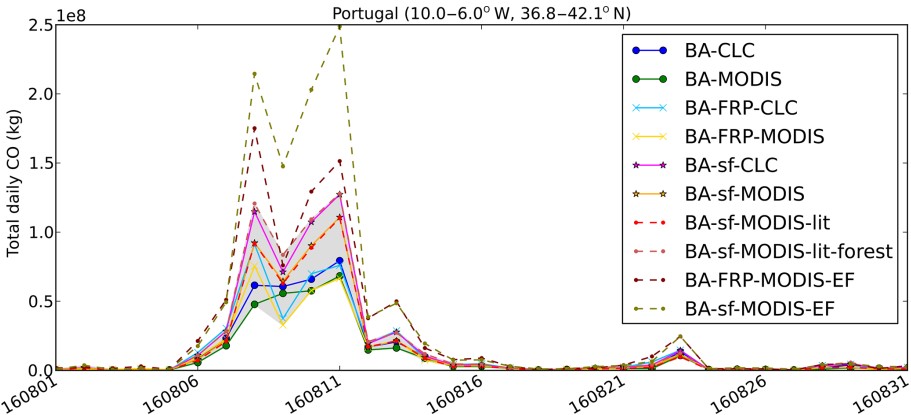

**Figure 6.** Total daily CO emissions during August 2016 over northern Portugal. Results using different APIFLAME configurations are shown in different colors (cf. Table 2). The shaded area shows the total spread for experiments modifying the dry matter burned (without impact of emission factors).

sites are considered in the statistical comparisons since the resolution chosen is not relevant for urban areas.

Satellite observations offer a good complement to surface in situ measurements since they provide daily observations over the full domain. Here, the total CO observations from the Infrared Atmospheric Sounding Interferometer (IASI) instrument (Clerbaux et al., 2009; George et al., 2009), carried on board the MetOp satellite series since December 2006, are used. IASI is a nadir-viewing infrared sounder with a swath of 2000 km, allowing global coverage twice daily (Equator crossing time 09:30 LST, ascending node) with a horizontal resolution of 12 km (at nadir).

In this study, the CO retrievals by the FORLI software (Hurtmans et al., 2012) for measurements on board the MetOp-A and MetOp-B platforms are used. Validation experiments against other satellite retrievals (George et al., 2009) and MOZAIC aircraft profiles (De Wachter et al., 2012) show an uncertainty lower than 10 % in the upper troposphere and lower than 20 % in the lower troposphere with a tendency to overestimate concentrations and to agree better with in situ data for daytime observations. For total CO, differences between IASI retrieval and other observations of

$\sim$ 7 % were obtained. The smoothing error associated with the vertically integrated viewing geometry, represented by the averaging kernels (matrix **A**), is particularly important for comparisons to model profiles. The vertical smoothing may be summarized as the number of degrees of freedom for signal (DOFS = trace(**A**)). For IASI CO, it varies between $\sim$ 0.8 and $\sim$ 2.4 depending on the surface temperature: a larger DOFS generally corresponds to a better sensitivity to lower vertical levels due to enhanced thermal contrast (typically warm continental surfaces). To maximize sensitivity to the surface, we have chosen to use the daytime data only (overpass around 10:00 UTC). Figure 7 shows examples of averaging kernels associated with CO partial column profile retrievals (18 layers, each 1 km, the last layer being 18 km to the top of the atmosphere) over Portugal and over the Atlantic Ocean during the summer of 2016. Over land, the maximum sensitivity is reached in the free troposphere, around 5–7 km higher than over the ocean. The lower sensitivity to the surface over the ocean is due to the lack of thermal contrast with the surface. These averaging kernels are applied to the model CO profiles for quantitative comparisons.

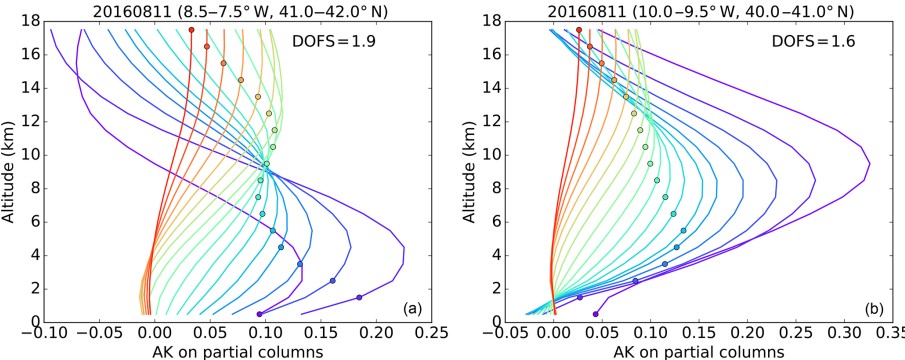

**Figure 7.** Averaging kernels (AKs) representing the vertical sensitivity of CO profiles retrieved from the IASI/MetOp-A daytime observations over land in Portugal **(a)** and the ocean off the coast of Portugal **(b)**.

For aerosols, the aerosol optical depth (AOD) at 550 nm and the 10 km horizontal resolution from the MODIS Collection 6.1 Level 2 products (MOD04_L2 for Terra, Equator crossing time 10:30 LST, and MYD04_L2 for Aqua, Equator crossing time 13:30 LST) (Levy et al., 2013; Levy and Hsu, 2015) are used. The combined dark target and deep-blue data are used, selecting only observations with a good to very good confidence level. The expected error is $\pm(0.05+0.15\,\text{AOD})$ for the dark target product and $\pm(0.03+0.20\,\text{AOD})$ for the deep-blue product. Sayer et al. (2014) also report good accuracy for the merge product compared to surface sun photometer data (AERONET network) over Europe (bias of $-0.01$, correlation of 0.86). Here, for consistency with the MetOp overpass time, only MODIS/Terra observations are used.

Observations from the Multi-angle Imaging Spectro-Radiometer (MISR) (Diner et al., 1998) on board the Terra satellite are also used to estimate fire plume height. The L2 products of wind-corrected stereo height (MIL2TCST, MISR_AM1_TC_STEREO) and cloud classifier (MIL2TCCL, MISR_AM1_TC_CLASSIFIERS), provided at 1.1 km resolution, were combined in order to only keep stereo height data corresponding to aerosols. Although most analyses of fire injection heights use plume-by-plume digitization with the MISR INteractive eXplore (MINX) software (Val Martin et al., 2010, 2018), the use of L2 retrievals has already shown good consistency with the MINX approach (Kahn et al., 2007; Mims et al., 2010). Fires are usually at their highest intensity in the afternoon (corresponding to the highest injection height). As the MISR observation is performed in the morning (Terra, Equator crossing time 10:30 LST), the fire plume height deduced will be quite conservative and will not correspond to a maximum.

The vertical distribution of aerosols is studied using observations from the CALIOP (Cloud-Aerosol Lidar with Orthogonal Polarization) instrument on board the Cloud-Aerosol Lidar Pathfinder Satellite Observation (CALIPSO) satellite (Winker et al., 2009). Here, the vertical feature mask (VFM) level 2 product v.4.20 (Winker, 2018) is used in order to identify the altitude of the aerosol layers and their dominant type (Kim et al., 2018). The classification includes two subtypes for the identification of aerosols from biomass burning: polluted continental and smoke, corresponding to non-depolarizing aerosols within the planetary boundary layer (PBL) and mixing both polluted continental aerosols and biomass burning aerosols (which have close optical properties), and elevated smoke, corresponding to layers with tops higher than 2.5 km, which may include non-smoke pollution lofted above the PBL.

### 5.3 CHIMERE-WRF regional CTM

The analysis is undertaken using the CHIMERE regional CTM (version 2017), driven by simulations from the WRF meteorological model version 3.7.1 (Skamarock et al., 2005) in its non-hydrostatic configuration. The parameterizations used in WRF for these simulations are mainly the same as those already used for studies over the Mediterranean area, such as Menut et al. (2016): the model reads global meteorological analyses from the National Centers for Environmental Prediction (NCEP) Global Forecast System (GFS) as large-scale forcing and uses spectral nudging (von Storch et al., 2000) to follow large-scale meteorological structures and to have its own structures within the boundary layer. Vertically, 28 levels are defined from the surface to 50 hPa. The single-moment five-class microphysics scheme is used, allowing for mixed-phase processes and supercooled water (Hong et al., 2004). The radiation scheme is a rapid radiative transfer model for global modeling (RRTMG) scheme with the Monte Carlo Independent Column Approximation (McICA) method of random cloud overlap (Mlawer et al., 1997). The surface layer scheme is based on Monin–Obukhov with Carlson–Boland viscous sub-layer. The surface physics is parameterized using the Noah land surface model scheme (Chen and Dudhia, 2001). The planetary boundary layer physics is estimated using the Yonsei University scheme (Hong et al., 2006), and the cumulus parameterization uses the ensemble scheme of Grell and Dévényi (2002). The

aerosol direct effect is taken into account using the Tegen et al. (1997) climatology.

Chemistry transport simulations with the CHIMERE model have been performed for the time period 1 June–31 August 2016 over western Europe with a horizontal resolution of 10 km and 20 hybrid vertical levels from the surface up to 200 hPa, using the MELCHIOR2 reduced gas-phase chemical scheme (44 species, almost 120 reactions) and the aerosol module by Couvidat et al. (2018) (including the aerosol microphysics, secondary aerosol formation mechanisms, aerosol thermodynamics, and deposition). The evolution of aerosol species (nitrates, sulfates, ammonium, primary organic matter (POM), secondary organic aerosol (SOA), elemental carbon (EC), marine aerosols, and mineral dust) is simulated using a sectional approach with 10 size bins (40 nm to 40 μm). The thermodynamic module ISOR-ROPIA v2.1 (Fountoukis and Nenes, 2007) is used for inorganic aerosols, and the module SOAP is used for organic aerosols (Couvidat and Sartelet, 2015). The optical properties of aerosols are calculated by the Fast-JX module version 7.0b (Bian and Prather, 2002) used in CHIMERE for the online calculation of the photolysis rates.

Initial and boundary conditions are derived from a 5-year (2004–2009) global reanalysis at a resolution of 1.125° from the MACC II project (Monitoring Atmospheric Composition and Climate II). The MACC modeling system relies on the coupled IFS-Mozart (Horowitz, 2003) modeling and assimilation system for reactive gases and on the MACC prognostic aerosol module for particulate matter (https://atmosphere.copernicus.eu/eqa-reports-global-services, last access: 24 June 2020).

Dust emissions are calculated following Menut et al. (2013b), biogenic emissions are calculated using the Model of Emissions and Gases and Aerosols from Nature (MEGAN) version 2.1 (Guenther et al., 2012), and sea-salt emissions are calculated using the Monahan et al. (1986) scheme. The anthropogenic emissions from the European Monitoring and Evaluation Program (EMEP) inventory are redistributed as described in Menut et al. (2013a). The biomass burning emissions from the APIFLAME model are included (described in Sect. 5.1). By default, emissions are assumed to be more intense during the day. The total daily emissions are thus redistributed over the day (the total remaining unchanged), assuming that 70 % of the total will be emitted during the day between 08:00 and 20:00 local time and the remaining 30 % at night.

In order to quantify the contribution from different sources to the simulated regional CO, CO tracers were included: CO from regional emissions by anthropogenic sources and biomass burning, secondary CO from chemistry, and CO from initial and boundary conditions. All are removed by reactions with OH. The sum of these five tracers is equal to the total CO.

A critical parameter for the simulation of the fire plumes is the injection profile of emissions. The plume rise model

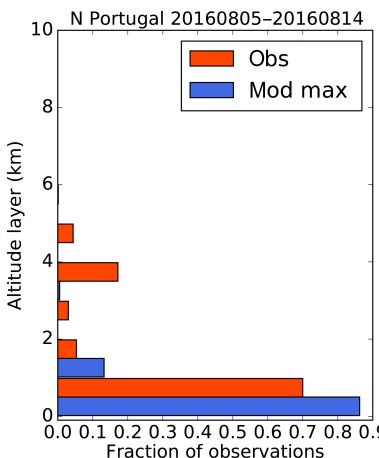

**Figure 8.** Distribution of the aerosol plume heights (1 km vertical layers) observed by MISR (obs) and simulated by CHIMERE at the same location and time (mod).

from Sofiev et al. (2012) is used, forced by the MODIS FRP as a surface constraint. The injection profile is derived by assuming homogeneous mixing below the maximum height. Menut et al. (2018) have tested different shapes of the injection profile in CHIMERE for the transport of biomass burning plumes in western Africa, which show very low impact on the simulated concentrations.

The vertical distribution of aerosol plume height observed by MISR above the fire region in Portugal in August 2016 is shown in Fig. 8. It corresponds to the number of detections on 1 km vertical layers for three overpasses on 7, 9, and 14 August (10, 3, and 176 observations, respectively). The coincident heights of maximum aerosol concentration simulated using the Sofiev plume rise model are also shown. While the aerosol layers remain below 2 km in the simulations, a significant fraction is located in the free troposphere according to MISR ($\sim 25$ % above 2 km). Due to the relatively low revisit time of the instrument, MISR can not provide a precise daily constraint on injection heights. The derived profile will be used to test the sensitivity of the simulations to the emission injection profile using a realistic distribution.

Lidar observations also allow the analysis of aerosol plume height and have been used in several studies to analyze injection height (e.g., Labonne et al., 2007). However, for this case study, there was no CALIPSO overpass above the fire region so there could be no constraint on the profile at the source of emissions. The classification data are used in Sect. 5.5.2 to evaluate the simulated altitude of transport of the biomass burning aerosols downwind from the emissions (Sect. 5.5.2).

In order to test the uncertainty associated with the various options on burned area processing models, several simulations were conducted, focusing on the largest influences on emissions: without fire emissions, using the emissions BA-FRP and BA-sf with MODIS vegetation, and using the

**Table 4.** CHIMERE sensitivity simulations. Burned area configurations are presented in Table 2.

| Name | Burned area | Plume rise scheme | MISR profile | SEVIRI diurnal |
|---|---|---|---|---|
| BA-FRP (default) | BA-FRP-MODIS | × | | |
| BA-sf | BA-sf-MODIS | × | | |
| BA-sf-MISR | BA-sf-MODIS | | × | |
| BA-sf-MISR-SEV | BA-sf-MODIS | | × | × |
| BA-sf-CLC | BA-sf-CLC | | × | × |
| BA-sf-EF | BA-sf-MODIS-EF | | × | × |

emissions forced by the local emission factors BA-FRP-EF and BA-sf-EF. In addition, the impact of injected height is tested using simulations with and without the vertical distribution observed by MISR. The impact of including the diurnal variability using SEVIRI is also tested. The simulations performed are summarized in Table 4. The simulations without fires and with the BA-FRP emissions were performed for the time period 1 June–31 August 2016, while other sensitivity simulations were performed starting on 5 August 2016 (using the restart file from the BA-FRP simulation).

## 5.4 Sensitivity of the simulated concentrations to the configuration of APIFLAME

Figure 9 shows the average surface CO concentrations during the summer of 2016 and the relative contribution of the different sources based on the tracer simulation (for the BA-FRP simulation). Due to its relatively long lifetime, CO is strongly influenced by boundary conditions over the whole domain. The simulation, started on 1 June 2016, shows a low influence from initial conditions. Chemical production (oxidation of volatile organic compounds; VOCs) increases the background levels by 10 %–15 %. Fire and anthropogenic emissions are dominant at the surface close to source regions. Fires in northern Portugal affect the whole country, and a large plume is transported towards the southwest over the Atlantic Ocean. On average over the summer, the contribution from fires to CO surface concentrations ranges from 66 % over the fire region to $\sim$ 10 %–20 % downwind over the ocean. For total CO (not shown), it decreases to 17 % maximum over the fire region to $\sim$ 3 % downwind over the ocean.

Three subregions will be discussed in more detail throughout this study: above the fire region in northern Portugal and in the fire plume outflow off the northern coast of Portugal and off southern Portugal. The CO and $PM_{10}$ speciation on average over the summer and the selected subregions during the fire event are shown in Fig. 10. On average over the summer, fire emissions increase surface CO by 22 % over the fire region and 10 %–12 % in the outflow over the Atlantic. During the fire event, these amounts increase to 63 % over the fire region and 50 % downwind. For surface PM including fire emissions, total concentrations increase by 50 % over the fire region on average during the summer (6 % downwind) and by a factor of 5 during the fire event (40 % down-

wind). The increase is mainly composed of organic carbon (OCAR) and other PPM (both have low contributions in the simulation without fire emissions). As explained in Sect. 3.4, the surrogate species "other PPM" (inert fine particles) is introduced to account for the missing mass of aerosols in the inventory (difference between the emission factor of $PM_{2.5}$ and the sum of emission factors for the identified aerosols). Majdi et al. (2019) have shown that its contribution to atmospheric concentrations of aerosols is of the same order of magnitude as the SOA produced by organic compounds of intermediate volatility and semi-volatile organic compounds (I–SVOCs). In the simulations presented here, the contribution of I–SVOCs to SOA formation is not included, and the fraction of SOA in the biomass burning plume is very low and most likely underestimated. For model versions including SOA formation from I–SVOCs, the contribution from the surrogate PPM species should not be taken into account in order to avoid double counting.

The average sensitivity of the surface and total CO and $PM_{10}$ to the factors influencing the biomass burning emissions tested in this study are reported in Table 3. The average differences are calculate for the fire plume over points with a contribution from fires > 10 %. The maps for CO are presented in Fig. 11.

As for the emissions, the largest impact is associated with the small fires (increased burned area) and the higher emission factors. For small fires, the relative difference is equal to 41 %–48 % on average in the denser part of the plume (relative contribution > 10 %). Using increased emission factors, the CO concentrations are more than doubled ($\times$2.5 at the surface, $\times$2.1 for the column). The impact on $PM_{10}$ is lower (lower increase in EF) but still almost as high as the contribution from small fires. Both effects are particularly marked for surface contributions. The influence of the different vegetation types is more nuanced. Using the CLC database results in an increase in CO in Portugal but not in other regions. The effect is lower than that of small fires and emission factors but still very significant, on average 10 %–30 %. These influences are in line with the sensitivity of emissions discussed in Sect. 5.1.

The sensitivity to injection heights is also presented, using either the MISR profile (resulting in $\sim$ 25 % of emissions injected above the PBL) or the default scheme in CHIMERE

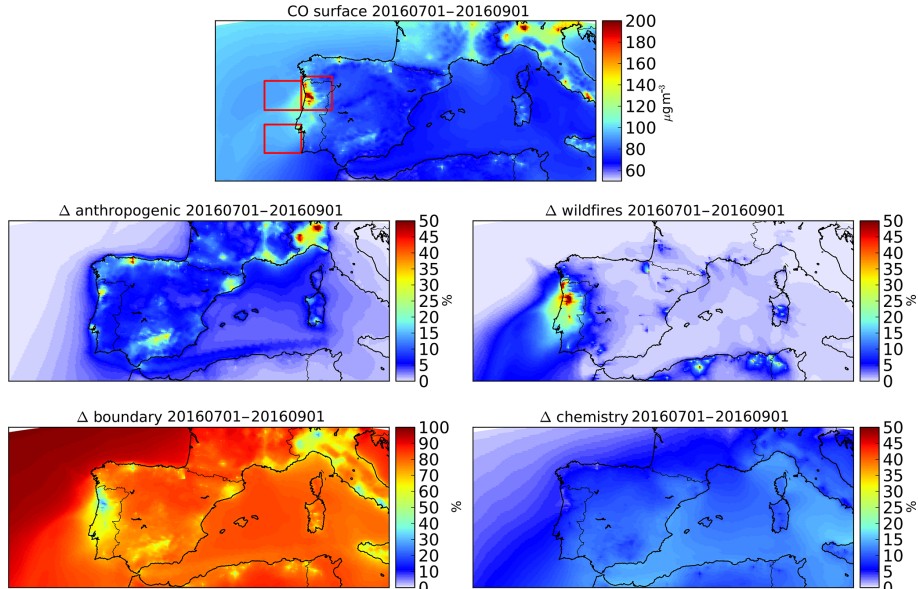

**Figure 9.** Average surface concentration of CO simulated by CHIMERE during July–August 2016 (top) and relative contributions from the main contributing CO tracers: from primary anthropogenic (anthropogenic) and biomass burning (wildfires) emissions, boundary conditions (boundary), and chemical production (chemistry). The red squares on the top map delimit the regions used for the evaluation of the fire plume simulation against satellite observations.

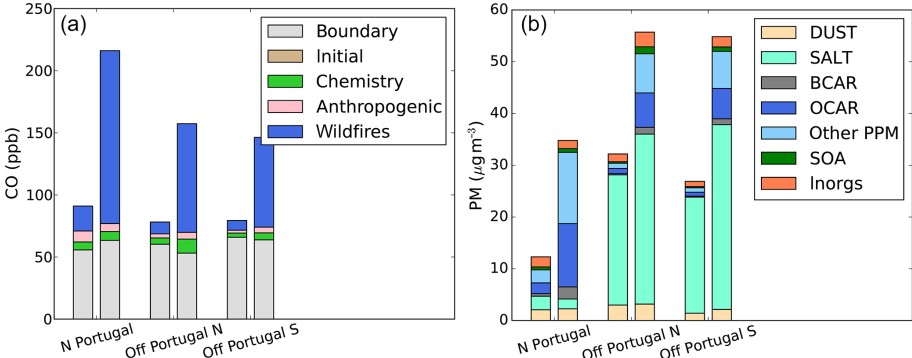

**Figure 10.** Average surface concentrations of CO (**a**) and PM$_{10}$ (**b**) over the subregions depicted in Fig. 9 (northern Portugal above the fires and off the coast of northern Portugal and off southern Portugal in the transported fire plume) with contributions from the different CO tracers and aerosol species: dust, sea salt (SALT), black carbon (BCAR), organic carbon (OCAR), secondary organic aerosols (SOA), and inorganic aerosols (Inorgs; including sulfate, nitrate, ammonium), and other primary particulate matter (Other PPM). For each region, the left bar is averaged over the whole summer, while the right bar corresponds to the intense fire episode at the beginning of August (8–14 August 2016).

(for this case, all emissions in the PBL). Having a fraction injected above the PBL mechanically decreases surface concentrations (by $\sim 25\,\%$). The total column tends to increase to the north of the fire region and decrease to the south. This shows that injection height has a significant impact on transport pathways. Northward transport (towards the Bay of Biscay) will tend to be in the free troposphere, while southward transport remains at low altitude.

The coefficient of variation across the sensitivity simulations (standard deviation / mean value), averaged during the fire event, is mapped in Fig. 12 for PM$_{10}$, PM$_{2.5}$, and CO. The variability is maximized by excluding the BA-

sf-SEVIRI-MISR experiment which shows little difference from the BA-sf-MISR experiment. For all considered compounds, variability is $\sim 30\,\%$ above fire regions and $\sim 20\,\%$–25\,\% over the other impacted areas of Portugal and reduces to $\sim 5\,\%$–10\,\% further downwind as the plumes are diluted. For comparison, Majdi et al. (2019) found a sensitivity of surface PM$_{2.5}$ to SOA formation from I–SVOC emissions of $\leq 30\,\%$ for the case study of the Greek fires during the summer of 2007. The choice of the processing methods of burned area can thus have as much impact as SOA production.

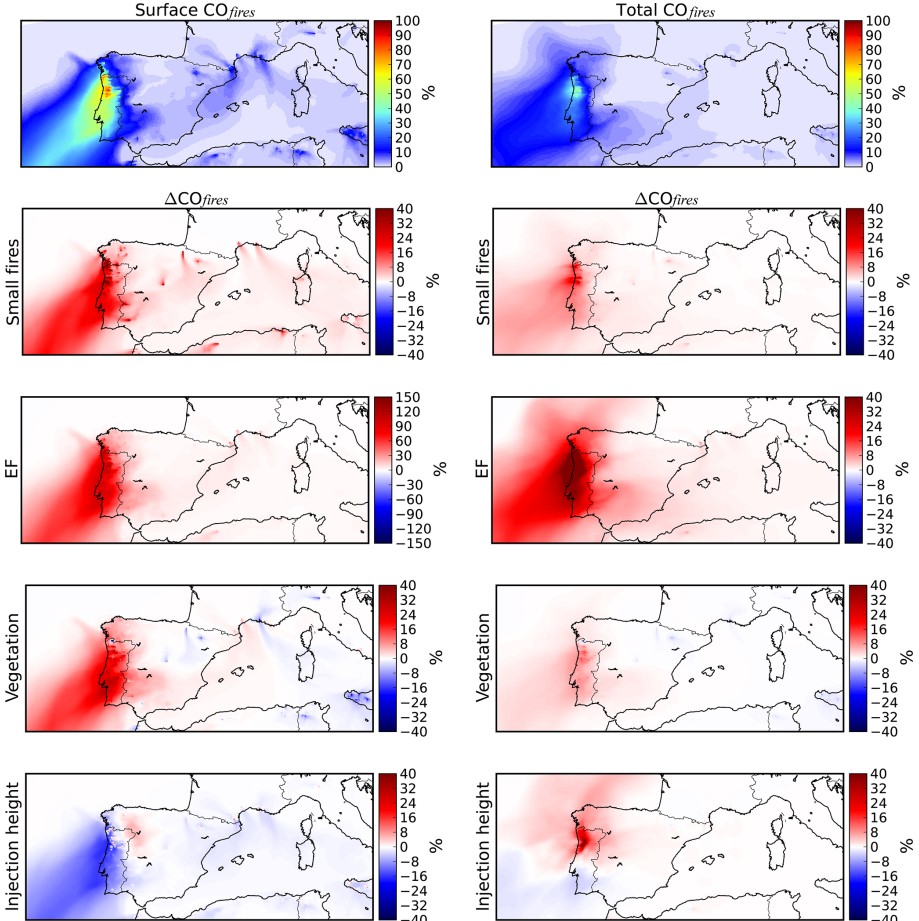

**Figure 11.** Top maps: relative impact of biomass burning emissions on surface concentrations (left) and total columns of CO (right) for simulation BA. Rows 2–5: sensitivity of surface (left) and total (right) CO to a given factor (*y* axis label), calculated using the difference between two sensitivity simulations as described in Table 3. Note the change in scale for the impact of EF on the surface concentration of CO.

## 5.5 Evaluation against observations

### 5.5.1 Surface observations

The surface concentrations of $PM_{10}$, $PM_{2.5}$, and CO simulated and observed during the largest fire event in Portugal (8–14 August 2016) are mapped in Fig. 12 (only rural background stations). The increase associated with biomass burning is significant in both observations and simulations. These maps show that background levels (PM in Spain, for example) are slightly underestimated and that the average concentrations in Portugal are overestimated at some stations and underestimated at others. For CO, the number of available measurements at rural and suburban stations is low especially in Portugal. Comparisons in Spain show large underestimates probably due to both underestimated background and underestimated local contributions. A strong impact from fires is observed in southern Portugal, as well as in the Lisbon area. However, the latter correspond to urban sites for which the resolution of the simulation may not be relevant, although

peaks during the fire event are consistent between observations and simulations (not shown).

Regionally averaged daily comparisons of $PM_{10}$ are shown in Fig. 13 for two subregions: northwestern (NW) and central western Portugal (CW), where most stations affected by the fire event are located. The simulated background level, before and after the fire event, is of the same order of magnitude as the observations, but the variability in the NW region is not well captured. This may be partly explained by missing long-range transport of dust from Africa, as discussed in Sect. 5.5.2.

During the fire event, both observations and simulations show two main increases with peaks consistent to within a day. For stations in the NW region, the first peak is overestimated in the simulations (and 1d too early), while the second peak is underestimated (and too late). Over the CW region, simulated concentrations are too high but also show stronger variability. It should be noted that the number of stations with available observations decreases during peaks. If the spread

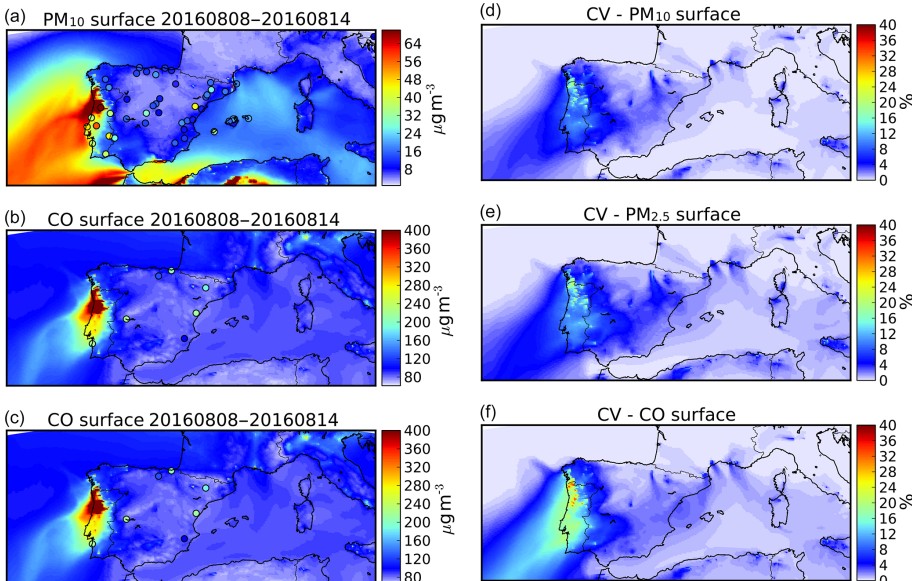

**Figure 12. (a–c)** Average surface PM$_{10}$, PM$_{2.5}$, and CO simulated by CHIMERE (experiment BA-sf-MISR) and observations at surface rural background sites (colored dots) during the time period 8–14 August 2016. **(d–e)** Coefficient of variation (standard deviation / mean value) of the ensemble of experiments (described in Table 4, excluding BA-sf-SEVIRI-MISR; cf. text for details).

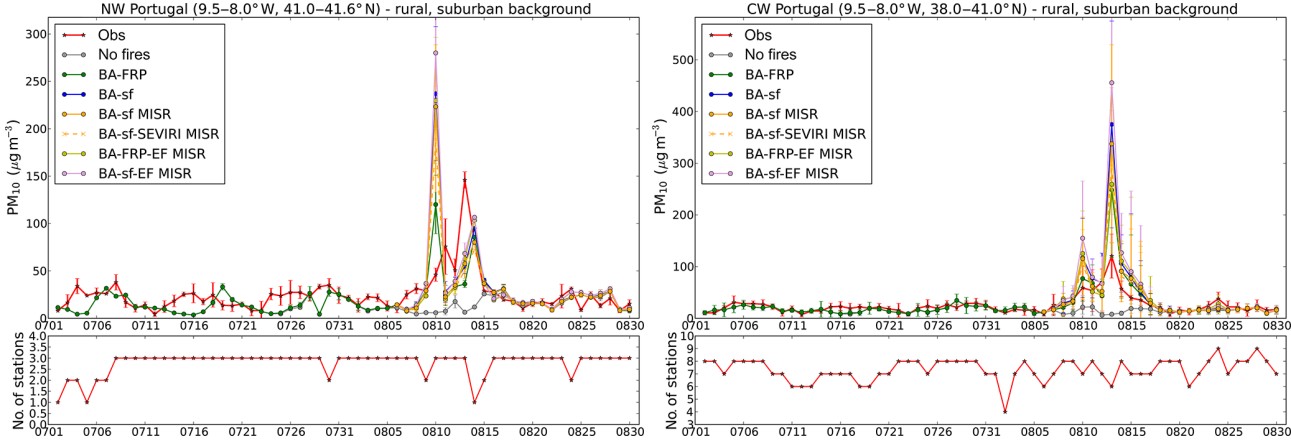

**Figure 13.** Regional average daily surface observations and CHIMERE simulations with different configurations (described in Table 4). The error bars correspond to the standard deviation of the daily observations across sites. The bottom plots show the number of sites included in the average for each day. The experiments "no fires" and BA-FRP were conducted for the whole summer and the sensitivity simulations for the time period 5–31 August 2016.

of the transported plumes is too large in the model, or if the temporal variability and transport is slightly shifted, some values associated with filtered peaks may increase values at a neighboring measurement site. The comparisons between observations and simulations for the simulation with lower contribution (BA-FRP) show an overestimate of 4 % on average over the 8–14 August time period for the NW region (72 % standard deviation) and 30 % on average over the CW region (46 % standard deviation). Accounting for small fire results in a strong increase (28 % and 33 % during 8–14 August for the NW and the CW regions, respectively) and thus

increases the overestimate compared to observations. Using the higher emission factors from Alves et al. (2011a) further increases surface concentrations (17 % and 27 % for the NW and the CW regions). Using the MISR vertical distribution only slightly decreases the peak values at these sites (11 % lower on average). Modeling a more precise diurnal variation using SEVIRI does not have a significant impact on these comparisons.

The generic emission factors and fuel consumption values used in this study are consistent with recent literature or are on the lower edge. The overestimate compared to surface ob-

servations may be due to an overestimated burned area or to uncertainty related to the integration of the emissions in the model and the transport of the resulting plume, such as an underestimated injection height, problems in the representation of the PBL by the WRF model, or a transport error.

### 5.5.2 Satellite observations of aerosols (MODIS and CALIOP)

The regional transport is further explored using comparisons with satellite observations. Figure 14 shows the comparisons with MODIS AOD during the main fire event (between 8 and 14 August 2016) for the simulation BA-sf-MISR. Only coincident and collocated values are compared so that comparisons will be affected by the total emissions but also by the transport error or a temporal shift in emissions. Note that the coverage is reduced by cloud cover (data filtered out) and depends on the satellite overpass. The maps thus represent a composite of the available observations during this time period rather than an average. It merges three main transport events: towards the Atlantic on 11 August, towards the south at the beginning of the fire event on 7–8 August, and towards the north (Bay of Biscay) on 13–14 August.

An underestimate of AOD is diagnosed with the model over the Atlantic and the Mediterranean, as well as over Spain and France. This underestimate may be due to a lack of emissions (anthropogenic, biogenic, fires, or mineral dust) or to the way AOD is calculated in the model (using the Fast-JX online model in CHIMERE). A classic candidate of this kind of underestimation in regional models and in summertime is a missing mineral dust plume coming from Africa. Gama et al. (2020) show that desert dust contributes significantly to PM concentrations in Portugal and are mixed with the fire contribution in 2016. The analysis of mineral dust concentrations and the Ångström exponent using global databases displayed with Giovanni (https://giovanni.gsfc. nasa.gov/giovanni/, last access: 27 February 2020, Fig. S2) confirms that there is a long-range transport of mineral dust above the Atlantic from Africa to Europe at the beginning of the fire event (6–9 August). It increases AOD above the Atlantic, northern Spain, and the Bay of Biscay on 8 August and southern Portugal on 9 August and will be mixed with the biomass burning contribution. However, there is no transport towards the Mediterranean. Since this is a background bias, homogeneous over land and ocean, and not a problem of plume with high values, this modeling problem has to be investigated more generally with the CHIMERE model, but it is not due to the biomass burning inventory presented in this study.

In regions affected by fires, simulated transport pathways are similar to observations with shifts in transport direction well reproduced. However, the intensity of the plumes is underestimated in the simulations especially downwind. Their horizontal spread is also larger, suggesting too much dispersion. These two elements could be partially explained by an injection height of fire emissions that is too low.

Daily comparisons over three subregions (Fig. 9) are shown in Fig. 15: above the area affected by wildfires and downwind off northern and southern Portugal.

As was already observed on the average maps, background levels tend to be underestimated in the simulations compared to observations. Comparisons in July and late August show that several peak values (e.g., around the middle and end of July and end of August) are underestimated. Here again, the mapping of dust with Giovanni suggests that these peaks are due to the long-range transport of dust. During the fire event, AOD increases in both observations and simulations between 7 and 15 August 2016 with peak values around 9–10 August and 11–13 August. Transport off southern Portugal is observed at the beginning of the event around 9 August and later around 13–14 August, while the transport off northern Portugal is mainly observed on 11–12 August. The observed peaks above the fire region and downwind are simulated at the right time but underestimated for all simulations, even using emissions including small fires that seemed to overestimate the burned area. Above the fire region, the simulations without small fires (BA) underestimate average values by 39 % (average over the time period 8–14 September). This reduces to 38 % if small fires are included and 14 % if the MISR profile is considered. The closest agreement is obtained if emission factors from Alves et al. (2011b) are used (2 % above the fire region). All simulations show underestimated AOD for the outflow from northern Portugal (16 %–38 % depending on the configuration). The closest agreement is obtained if emission factors from Alves et al. (2011b) are used (2 % above the fire region, 16 % off northern Portugal and 12 % off southern Portugal). For this case study and considering the available observations, using the diurnal cycle as derived from SEVIRI does not result in significant differences. Using the average vertical injection profile from MISR allows better agreement above all regions.

In order to test the possible impact of a transport error, the daily regional average and maximum simulated AOD are included in the comparisons (shaded area), together with the values collocated with MODIS observations. The maximum values are closer to the observed AOD or significantly higher. A bad timing in emissions and a small shift in transport may explain part of the underestimate. Another explanation is, here again, a missing inflow of dust from long-range transport especially at the beginning of the event. A transport of dust in the free troposphere would also explain that the underestimate obtained for AOD was not obtained for surface $PM_{10}$ (which was, on the contrary, overestimated).

The altitude of the aerosol plumes is analyzed using the aerosol layer classification from CALIOP (VFM product). Four CALIPSO overpasses are available during the studied time period:

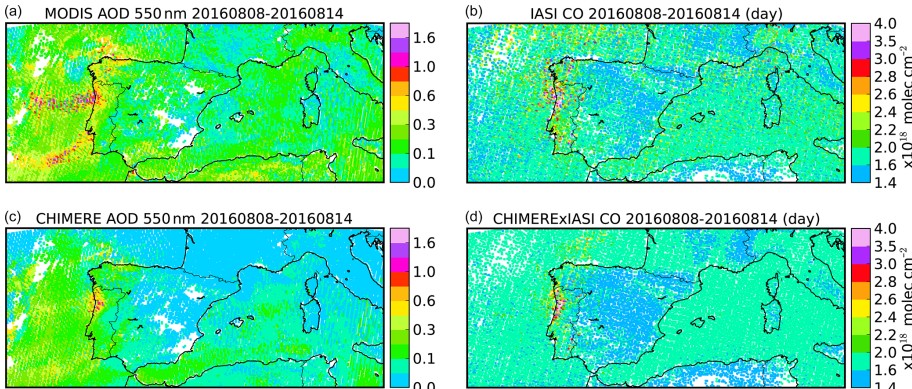

**Figure 14. (a–b)** Observations of AOD at 550 nm by MODIS and CO total column by IASI during the fire event in Portugal between 8 and 14 August 2016, averaged onto the CHIMERE grid at 10 km horizontal resolution. **(c–d)** Corresponding CHIMERE values (smoothed by IASI averaging kernels for CO) for the BA-sf-MISR experiment.

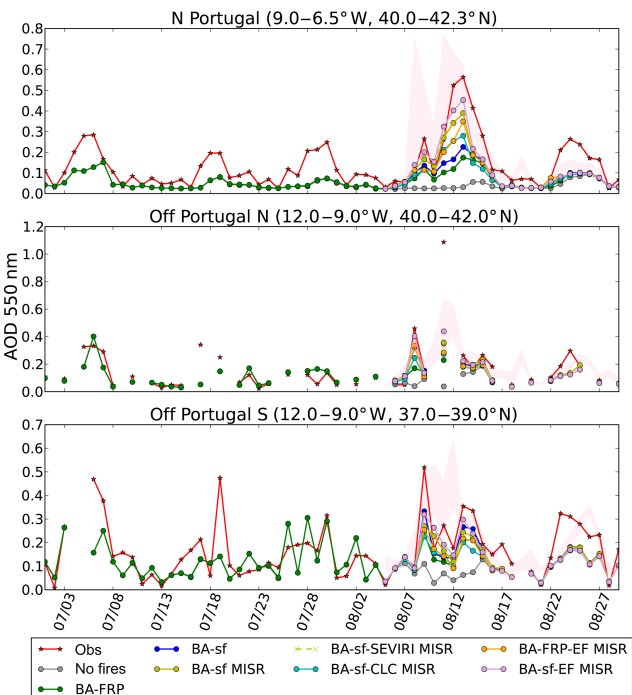

**Figure 15.** Daily average observed (MODIS) and simulated (CHIMERE) AOD at 550 nm during July–August 2016 over three subregions (mapped in Fig. 9). Results from different CHIMERE simulations are plotted (described in Table 4). The spread between the simulated regional average and maximum without collocation with MODIS data is also shown (shaded red area).

– 8 August, 02:44 and 13:24 UTC, both located above the Atlantic and capturing the outflow from the fires in northern Portugal;

– 10 August, 13:11 UTC, the only one above the continent but to the east of the fire region, capturing the recirculation above southwestern Spain;

– 15 August, 02:50 UTC, located above the Atlantic, capturing the outflow to the north of the domain but at the edge of the domain so that part of the plume is lost in the simulations.

The observed VFM on 8 August and the simulated $PM_{10}$ concentrations along the CALIPSO track are shown in Fig. 16 for the nighttime overpass and Fig. 17 for the daytime overpass. A large contribution from dust is observed in the free troposphere (2–7 km) at latitudes > 42° N, confirming that AOD at the beginning of the fire event corresponds to a mix of smoke and dust, the latter being underestimated in the simulations. The large contribution from fires simulated around 42° N up to ∼ 4 km is attributed to clean marine aerosols in the VFM (due to the color ratio). The 1064 nm backscatter is high and may actually correspond to the fresh smoke outflow. It is observed with similar structure but at slightly lower altitude by CALIOP. The smoke contribution in the southern part of the plume is observed at higher altitude with CALIOP (up to 3 km) compared to the model (main contribution below 2 km). For the daytime overpass, the observed plume is mainly attributed to dust and probably corresponds to mixed dust and smoke contributions. The model simulates a large contribution from fires around 3– 4 km, which is consistent with the observations only if the MISR profile is used. On 10 August, the tropospheric smoke aerosols are observed up to 2.5 km around 38° N and up to 3.5 km further north. For this case, the simulated layer height is in better agreement without the MISR profile although slightly higher (up to 4 km without MISR, 5 km with MISR) around 39° N. On 15 August, CALIOP captures the plume at ∼ 1 to 4.5 km. Using the MISR profile in the simulations increases this transport pathway (Fig. 11). However, the plume is simulated too high using the MISR profile (1 to 6 km) but too low without MISR (∼ 3.5 km). These comparisons highlight the difficulty in simulating the fire plume height but also

highlight that using an averaged profile from MISR is a good option if no other observation is available.

### 5.5.3 Satellite observations of CO (IASI)

Figure 14 shows the comparisons between the IASI total CO retrieval and the different simulations during the main fire event (between 8 and 14 August 2016) for the simulation BA-sf-MISR. Compared to observations, the simulated background total CO levels are too low particularly over Spain. As for AOD, this could be due to missing local sources or to an underestimated contribution from long-range transport. Current global models also tend to underestimate CO levels in the Northern Hemisphere during the summer by $\sim 10\%$ (e.g., Monks et al., 2015) even for simulations specific to the studied time period. Here again, we will concentrate on the increase above the background level to evaluate the simulated signature from wildfires.

Daily comparisons over the three subregions (Fig. 9) are shown in Fig. 18. The simulated total columns with and without smoothing by the IASI averaging kernels are shown. The comparisons during periods not affected by the fire event highlight a slight underestimate of the simulations in July above fires ($-1\%$) and overestimate over the ocean ($\sim 3\%$). At the end of August, simulations are lower on average by $-8\%$ above fires and $\sim -3\%$ over the ocean. These differences remain lower than the expected uncertainty of IASI total CO retrieval ($\sim 7\%$).

The strong increase during the fire event is also clearly observed by IASI with the same daily variation and transport pathways as observed by MODIS: peak around 9–10 August and then 11–13 August. Total CO values are underestimated by 18% on average over the fire region and off northern Portugal, 16% if small fires are included, and 8% if the MISR profile is used. If the emission factor from Alves et al. (2011b) is used, the simulated total CO becomes overestimated over the fire region if small fires are included (16%) and is in good agreement for the BA-FRP-EF simulation (2% average difference). Above southern Portugal, simulations are underestimated by 2%–6% (minimum difference using adjusted emission factors and maximum difference without the MISR profile), but the peak value on 10 August is strongly underestimated ($\sim 20\%$).

For CO, the difference between the values simulated with and without the MISR profile is particularly marked due to the smoothing by the averaging kernels, which peak in the free troposphere for IASI. Figure 19 shows average CO profiles over the three regions considered and for 2 d: 8 August at the beginning of the fire event and 11 August between the peaks. Above the fire region, the observed and simulated profiles show a peak at 2–3 km and lower values towards the surface. On 8 August, this shape is accentuated in the model after applying the averaging kernel even for the simulation with emissions mixed in the boundary layer, suggesting that it may in part be explained by the sensitivity of the observing system (observation and retrieval process). Injecting emissions higher (simulations using the MISR vertical profile) results in a better agreement everywhere and particularly above fires, again at least in part due to the shape of the IASI averaging kernels. Simulations tend to overestimate CO above the fire region on 8 August (especially for the BA-FRP-EF experiment) but underestimate on 11 August when the transported plumes are strongly underestimated particularly for the strong outflow on 11 August off northern Portugal. For this case, smoothing by the averaging kernel sharply decreases the simulated concentration peaks which are located in the lower troposphere, where observations above the ocean show little sensitivity (as shown by the shape of the averaging kernels in Fig. 7). For comparisons off southern Portugal, using the MISR injection profile reduces the CO concentrations on 8 August. This is consistent with a transport at low altitude in the model highlighted by the comparisons with CALIOP and with a decrease in total CO over the southern plume when emissions are injected at higher altitude (Fig. 11).

This comparison shows that the regional contribution is simulated with a good temporal variability and order of magnitude. The uncertainty of the plume heights makes the evaluation of emissions difficult since IASI is not only sensitive to the amount of CO but also to the altitude of transport.

## 6 Summary and conclusions

The APIFLAME biomass burning emissions model allows the calculation of aerosol and trace gas emissions based on observed burned area. The current version of the model (v2.0) uses the MODIS Collection 6 fire products of burned scars (MCD64A1, providing the date of burning) at 500 m resolution and active fires (MOD14, including hotspot detection and associated FRP) at 1 km resolution. For each fire detected, the vegetation type burned is attributed using the MODIS annual vegetation cover product (MCD12Q1) or the CORINE Land Cover (CLC) or USGS land use databases. The corresponding fuel consumed is derived from either OR-CHIDEE land model simulations or tabulated values from the literature. The carbon consumed is converted to trace gas and aerosol emissions for a list of species for which emission factors are available. Emission fluxes for model species are then derived using an aggregation matrix.

APIFLAME may be used for near real time applications (using MOD14 only). Forecasting the evolution of the emissions remains uncertain. However, the likelihood of a fire being controlled and extinguished increases when weather conditions are less favorable for its spread. A possibility is to modulate emissions with forecasts of fire weather indices computed from the forecasted meteorology, as suggested by Di Giuseppe et al. (2017).

APIFLAME was constructed to be modular in terms of input datasets and processing methods. In addition to the dif-

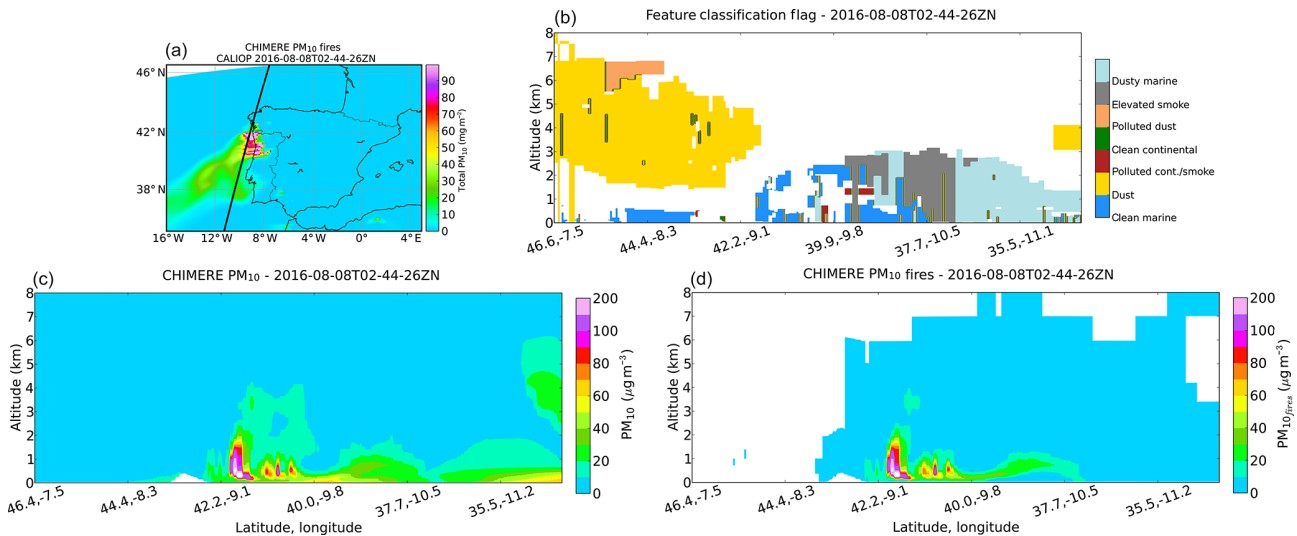

**Figure 16. (a, b)** Map of the CALIPSO nighttime overpass track on 8 August (in black) above the simulated total PM$_{10}$ mass density from fires (same time) and corresponding CALIOP vertical feature mask (VFM). **(c, d)** Vertical distribution along the same track of total PM$_{10}$ **(c)** and PM$_{10}$ from wildfires **(d)** from simulations (BA-sf MISR).

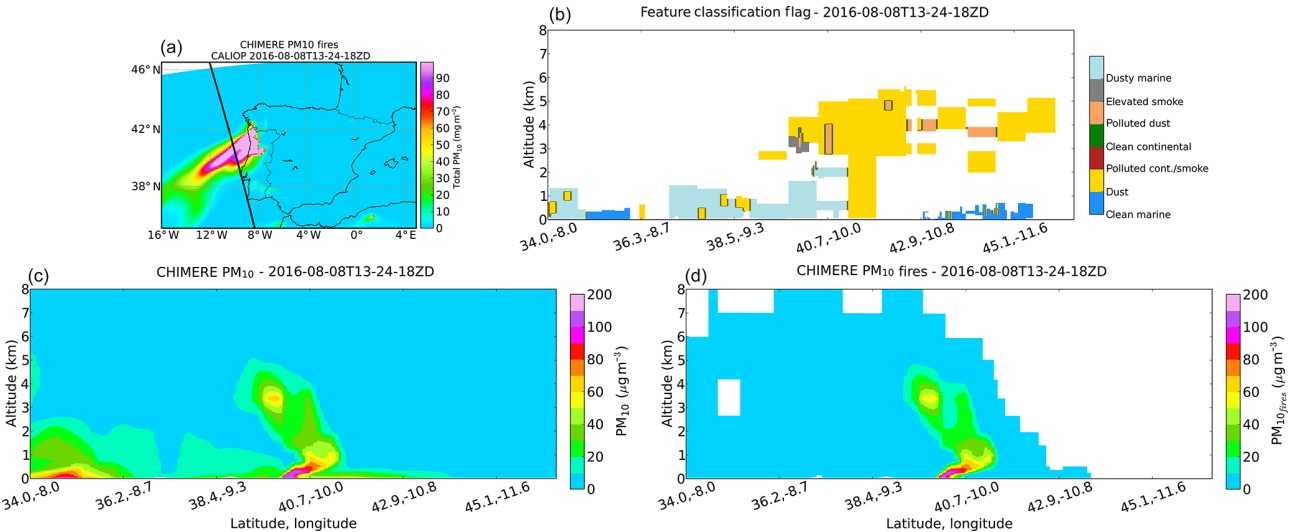

**Figure 17.** Same as Fig. 16 but for the daytime overpass.

ferent vegetation databases and the possibility of modifying emission factors as input parameter, different options for the calculation of burned area may be chosen. The main evolution in v2.0 is the possibility of merging burned area and FRP observations. Users may chose (1) to use the burned area calculation based on the MCD64A1 product only, (2) to redistribute the total monthly BA using the daily FRP value within each grid cell (BA-FRP option), or (3) to use the BA product but add active fires that are not collocated with a burned scar (that may correspond to small fires) with a modulation based on the FRP (BA-sf option). In addition, a diurnal profile may be applied to the daily emission fluxes using the geostationary observation of FRP (scaled) by SEVIRI (for Europe and

Africa). This does not change total daily emissions. Including small fires significantly increases the burned area. On average over the 2013–2017 period, it increases by $\simeq 46\%$ in boreal and temperate North America; 37% in equatorial Asia; 18%–22% in Europe, southeast and boreal Asia, and Central and South America; and 5% in Africa and Australia. These values are lower than the small fire contribution estimated by Randerson et al. (2012) for the MODIS Collection 5 fire products. This directly increases emissions. For CO, for example, the increase ranges from 5% in Africa and Australia to 60% in North America.

The ability of the model to provide useful information for the simulation of the impact of biomass burning on air quality

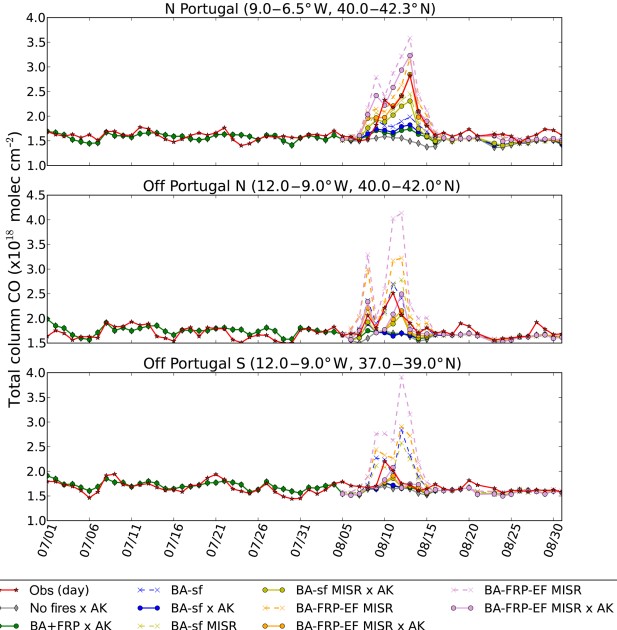

**Figure 18.** Daily averaged observed (IASI) and simulated (CHIMERE) CO total columns during July–August 2016 over three subregions (mapped in Fig. 9). Results from different CHIMERE simulations are plotted (described in Table 4). Simulated total CO are plotted with and without smoothing by the IASI averaging kernels (solid lines, reference "× AK", and dashed lines, respectively).

using a CTM is illustrated here for the case of the forest fires in Portugal during the summer of 2016 using the CHIMERE CTM. Depending on the burned area processing method, the total burned area ranges from 99 849 ha (MCD64 burned area product) to 144 882 ha (including small fires), while EFFIS reports that 115 788 ha burned. In our method, small fires are included depending on the FRP so that it can represent both the spread and the intensity of burning. The modularity of APIFLAME was used to provide information on uncertainty of the calculated emissions. Several key parameters of the model are tested: burned area, vegetation database, fuel consumption, and emission factor. Emissions with different processing methods of the burned area are used in order to analyze the impact on simulated concentrations: burned area only (BA), FRP daily variability (BA-FRP merged approach), and including small fires (BA-sf). The impact of attribution of the vegetation burned is tested using either the CLC or MODIS IGBP vegetation classification. The impact of the fuel consumption is tested using either that calculated from ORCHIDEE land model simulations or tabulated values. Finally, emission factors from average values for the fire type or using specific values for the region studied are used. Using the ensemble of emissions calculated, the average variability over 10 km grid cells and at daily resolutions during the event is 80 % for CO emissions, 60 % without considering small fires (variability due to the vegetation database, the

daily variability of the burned area, and the calculation of the fuel consumed), and 76 % without experiments on emission factors. Accounting for small fires significantly increases the total emissions during the event (33 %–36 %). The other critical parameter is the emission factor used. For this case study, the values reported in the literature for the specific vegetation burned are very different from the average values for the fire type/ecozone. In their intercomparison of different inventories, Carter et al. (2020) find a low impact of emission factors because all inventories considered use average values for different ecozones. For the case of the fires in Portugal, the impact of the fuel consumption is low, but the impact of the vegetation database can be significant (±30 % on average). It is not similar for all species due to the different emission factors.

The resulting impact on surface CO and PM concentrations, as well as total column CO and AOD, is simulated using the CHIMERE CTM driven by the WRF meteorology. The sensitivity to the different configurations of APIFLAME is tested, as well as the sensitivity to the injection profile used. Therefore, different vertical injection profiles have been used: calculated in CHIMERE based on the FRP (all below 2 km) and using an averaged vertical distribution derived from MISR plume height observations ($\sim 25$ % above 2 km). Over the fire region, the fire emissions contribute to 22 % of surface CO and 50 % of surface PM on average over the summer. During the fire event, they become the dominant regional source (63 % increase in surface CO, a factor of 5 for surface PM) with also a significant impact downwind. The variability across experiments is $\sim 30$ % over the fire region, 5 %–25 % downwind, and decreasing as the plume dilutes. As for the emissions, the largest impact is related to the burned area calculation (with or without small fires) and to the choice of emission factor. The modification of the injection profile directly impacts surface concentrations ($\sim 25$ % of emissions injected above the PBL results in $\sim 25$ % lower surface concentration) but also the total columns due to a modification of the transport pathways.

The different simulations performed are compared to the available surface and satellite observations. Since limited area simulations are performed, both CO and PM are affected by an inflow which depends on the boundary conditions chosen. The purpose of this paper being the evaluation of a specific fire event, climatological boundary conditions were chosen, but this could result in an underestimate of background levels compared to observations. Comparisons to observations show that background levels are $\sim 3$ % too low for total CO compared to IASI and $\sim 6$ %–30 % for AOD compared to MODIS. For the MODIS AOD, this is at least in part due to the contribution of the long-range transport of desert dust from northern Africa.

Comparisons with both surface and satellite observations show that the increase in concentration from the Portuguese wildfires is simulated at the right time but that it is difficult to have the peak values with good temporal variability (±1 d

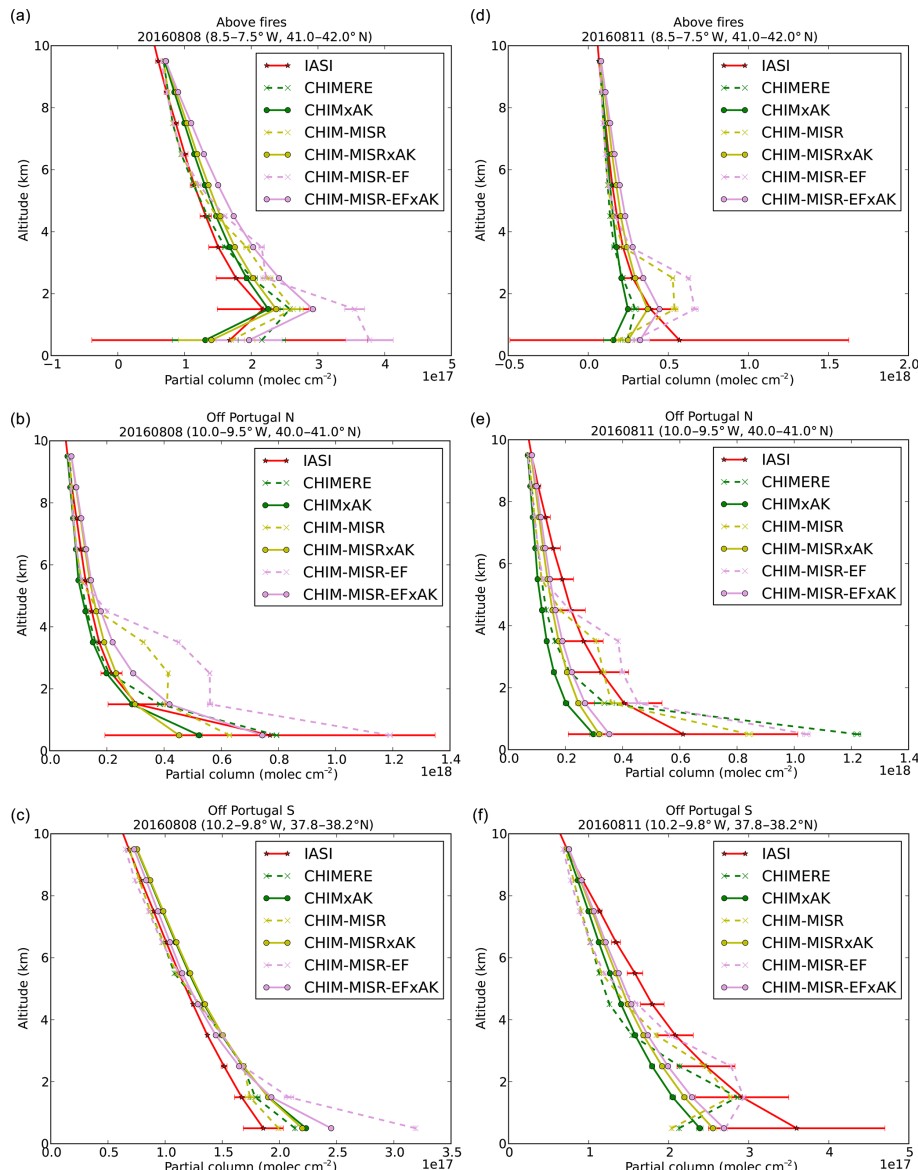

**Figure 19.** CO partial column profiles retrieved from IASI observations averaged over three subregions (mapped in Fig. 9) on 8 August 2016 **(a–c)** and 11 August 2016 **(d–f)** and collocated CHIMERE profiles with (solid line) and without (dashed line) smoothing by the IASI averaging kernel for three experiments: BA-sf (CHIM, green), BA-sf with the MISR injection profile (CHIM-MISR, yellow), and BA-FRP-EF with the MISR injection profile (CHIM-MISR-EF, pink). Profiles are only plotted up to 10 km for clarity. Error bars correspond to the standard deviation of data used in the average, corresponding to the variability within the chosen regions.

usually for the regional total). The lack of surface observations does not allow a statistically significant comparison for CO. For PM, comparisons to observations from three stations in northwestern Portugal closer to the fires and nine stations in Central Portugal further downwind show that the simulations overestimate concentrations (4 % in NW region, 30 % in CW) especially when small fires are included (increase of ∼ 30 %). Using higher emission factors from reported local observations further increases the differences. Using the MISR vertical distribution for emissions results in a small decrease at these stations (11 %), and accounting for the di-

urnal variability as observed by SEVIRI has a small impact on the daily comparisons. A similar performance was obtained in past analyses with APIFLAME for Europe (Majdi et al., 2019) and by Jose et al. (2017), who used other inventories for the analysis of Portuguese fires in 2010 with the WRF-Chem model ($PM_{10}$ overestimated by about 20 % in their best configuration). Part of the overestimate could be related to the use of the other PPM surrogate species, which accounts for a significant fraction of surface PM. Majdi et al. (2019) found that the contribution from this surrogate species to the fire-related $PM_{10}$ could be linked to SOA formation.

SOA concentration is very low in the CHIMERE simulations considered here and are probably underestimated. However, the contribution from other PPM may be overestimated in this study. Errors in the temporal variability of emissions and their injection height, as well as on their transport and spread, could also result in strong biases in the comparisons.

A larger-scale evaluation is allowed through the comparison to satellite observations. Unlike comparisons to surface data, these suggest an underestimate of the contribution from fires. Best agreement, with an underestimate of 5 % for AOD and 8 % for total CO above the fire region, is obtained when small fires and the MISR vertical distribution are considered. The underestimate, especially for CO, is larger in the outflow over the ocean and especially off southern Portugal. The experiment with increased emission factors significantly increases total CO and AOD. Total CO becomes too large over the fire region if small fires are included (16 % too high on average), while a good agreement is obtained using the merged BA-FRP approach (2 % average difference).

The apparent conflict in the conclusions of the comparisons to surface observations (overestimated peaks) or satellite observations (tendency to underestimate vertically integrated values) has been found in several studies (e.g., Majdi et al., 2019; Carter et al., 2020) and may have several origins. First, the representativity of the surface data is low, and transport errors will thus have a very strong impact on comparisons. Secondly, transport may be simulated too low or may be too wide on the vertical (due to the vertical resolution and numerical diffusion). The use of MISR plume height observations clearly improves comparisons, as already obtained in other model experiments (Rea et al., 2016; Zhu et al., 2018), but the lack of horizontal and temporal coverage of the instrument does not allow enough variability to represent the strong influence of fire intensity on injection heights. Comparisons to aerosol vertical structures observed by CALIOP show that the plume is transported at too low an altitude towards the south (and at about the right altitude closer to the fire region). A vertical dispersion that is too low mechanically results in surface concentrations that are too high. For AOD, a desert dust transport event most probably contributes significantly to the observed AOD and is not simulated in this study (the domain does not include northern Africa).

For CO, the underestimate of the transported plumes is at least partly due to the altitude of the transport, which is very critical for comparisons to IASI observations. Indeed, IASI measurements are primarily sensitive to the free troposphere above oceans. Despite the limitations due to the lack of in situ measurements over the region and in general close to large fire events, the good spatial and temporal coverage of these satellite observations provide very helpful information on the emissions' spatial and temporal variability.

Another issue for the simulation of biomass burning plumes is the dilution being too fast, a common problem for Eulerian models due to the fast dissipation of the transport scheme (e.g., Mailler et al., 2016), which may in part be due

to the vertical resolution in the free troposphere according to the analysis of Eastham and Jacob (2017).

As a conclusion, in spite of the large uncertainty of emissions, the case study analysis shows that the use of fire emissions derived from satellite observations of fire activity allows the attribution of the events to wildfires with correct timing (simulation of the peak values at ±1 d) and the estimation of their impact on surface concentrations with correct orders of magnitude. The modularity of APIFLAME allows the generation of ensemble emissions which provide information on uncertainties.

For chemistry-transport modeling applications, our recommendation is to compute emissions using both the BA-FRP and the BA-sf configurations in order to estimate the contribution from possible missing small fires. When available, it should be combined with observed vertical distribution (MISR here) in order to estimate their injection profile and avoid overestimating the impact at the surface. The use of emission factors reported for a vegetation type as close as possible to the vegetation burned is also recommended, although it is often impossible. The emission factors database used in the model will be updated regularly as new information becomes available (e.g., Andreae, 2019) particularly with the results of ongoing experiments in the US.

*Code availability.* The APIFLAME v2.0 biomass burning emissions model and associated documentation are available for download at https://doi.org/10.14768/20190913001.1 (Turquety et al., 2019). The global burned area derived from the MODIS satellite observations for the period 2014–2017 are available at https://doi.org/10.14768/20190913002.1 (Turquety, 2019a), and the files corresponding to the case study in southern Europe during the summer of 2016 are available at https://doi.org/10.14768/20190913003.1 (Turquety, 2019b). APIFLAME is a model under constant development, and the latest version, as well as its documentation and a test case, is available at http://www.lmd.polytechnique.fr/chimere/CW-fires.php (last access: 24 June 2020).

*Supplement.* The supplement related to this article is available online at: https://doi.org/10.5194/gmd-13-1-2020-supplement.

*Author contributions.* ST is the main developer of the APIFLAME model, ran the APIFLAME and CHIMERE simulations, wrote most of the paper, and produced all figures. LM and GS contributed to the APIFLAME model development. LM performed the WRF simulations and wrote the description in Sect. 5.3. LM, GS, SM, and ST contributed to the CHIMERE model development, including the integration of fire emissions. JHL, MG, DH, PFC, and CC contributed to the analysis of the IASI observations, including data preparation and comparisons to model simulation outputs.

*Competing interests.* The authors declare that they have no conflict of interest.

*Acknowledgement.* IASI is a joint mission of EUMETSAT and CNES. The authors acknowledge the AERIS data infrastructure for providing access to the IASI data in this study and the ULB-LATMOS for the development of the retrieval algorithms. The MODIS MCD64 and MOD14 products were retrieved from the NASA EOSDIS Land Processes Distributed Active Archive Center (LP DAAC), USGS Earth Resources Observation and Science (EROS) Center, Sioux Falls, South Dakota (https://lpdaac.usgs.gov/products/mcd64a1v006/, last access: 24 June 2020, and https://lpdaac.usgs.gov/products/mod14v006/, last access: 24 June 2020). The MODIS aerosol product MOD04 version 6.0 was retrieved from the online archive, courtesy of NASA's EOSDIS Level 1 and Atmosphere Archive and Distribution System (LAADS) Distributed Active Archive Center (DAAC) of the NASA Goddard Space Flight Center (GSFC) (https://ladsweb.modaps.eosdis.nasa.gov/archive/allData/61/MOD04_L2, last access: 24 June 2020). The CALIPSO Lidar Level 2 vertical feature mask (v4-20) was obtained from NASA Langley Research Center's Atmospheric Sciences Data Center. The global simulations for the construction of the initial and boundary conditions for CHIMERE were provided by the MACC-II project, which is funded through the European Union Framework 7 program. It is based on the MACC-II reanalysis for atmospheric composition; full access to and more information about the data can be obtained through the MACC-II website https://atmosphere.copernicus.eu/, last access: 24 June 2020).

*Financial support.* This research has been supported by the MISTRALS program (ADEME, CEA, INSU, and Meteo-France), as part of the ChArMEx project, for the development of APIFLAME, and by the Centre National d'Etudes Spatiales (CNES, France) for the analysis of IASI observations.

*Review statement.* This paper was edited by Gerd A. Folberth and reviewed by two anonymous referees.

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

**Remarks from the language copy-editor**

CE1  Could you please verify the changes made to this column? I inserted some semi-colons and I don't want to affect your meaning.