# Peer review of "APIFLAME v2.0 biomass burning emissions model: impact of refined input parameters on atmospheric concentration in Portugal in summer 2016"

_Geoscientific Model Development, 2019_

## Referee Comment (RC1) · Anonymous Referee #1 · 7 Nov 2019

This study presents an assessment of the APIFLAM v2.0 biomass burning and emissions model over a case study of forest fires that occurred in Portugal, 2016. It compares the modelled results from burnt area and emissions of Carbon Monoxide and aerosol particle matter (PPM) with different observed datasets and quantifies the uncertainties with a set of sensitivity experiments. There are several major issues with this paper, and major revisions are needed before the manuscript can be published.

1) The way the manuscript is written, and structure is very confusing, making it hard to read through. I would recommend this to be reviewed by an English native speaker and I would advise restructuring the text in some of the sections in order to improve readability and conclusions.

2) Section 4 shows the methodology for merging burned area with radiative fire power (RDP) data in order to increase the temporal variability which is crucial for estimating emissions for this study. It is shown how the RDP of small fires contributes to an increase of the burnt areas estimates. It would add value to this analysis to include an assessment on how this could impact fire emissions (e.g. using datasets such as GFAS).

3) It is mentioned in L288-289 that for MIDIS IGBP "to limit uncertainties in APIFLAME, fires detected in areas of chaparral or mediterranean vegetation types and classified in shrubland but also savanna are attributed to chaparral". Further along in L295-298 the authors state that for CLC forcing the biome of temperate forest allows a better agreement between the results of the experiments. Why not use a consistent approach and use the different datasets to understand the sensitivity to uncertainties regarding the land surface categorization?

4) In section 5.4 it is said that Secondary Organic Aerosols (SOA) have a very low fraction and can be discarded as a justification for not accounting for these. However, this is later presented as a limitation of this study affecting the bias of the modelling approach.

5) This study uses climatologies as boundary conditions for the chemistry and aerosol species. This could have a significant impact on the background levels of these species and contribute to the bias presented away from fire regions. Considering this, what would be the impact of using real time boundary conditions in the conclusions here presented? This is only pointed as a limitation of this study.

6) In the paragraph between lines 440 and 457 it is mentioned that regional fire emission reports from Alves et al. (2011) are larger than the ones used in this study. Considering that there are specific emission factors in the literature for the study region why was not do an assessment of the sensitivity of these fire emissions compared to the values proposed by other authors and the ones used in this work? Would the conclusions of this study be affected by the choice of the emission factors and land cover database? These are topics that are mentioned in this paper but are not explored, which would add value to this paper.

7) It is stated that if not explained by excessive emission factors, the overestimate of CO concentrations in simulations can be explain by other factors such as the temporal variability of emissions, problems in representing the Planetary Boundary Layer or injection of fire burning plumes. Do these factors have a greater influence than the uncertainty in the plant functional types? This is especially relevant since the plant functional type determines the emission factor and the fuel load available for emissions. Having an analysis of these contribution would significantly increase the value of the results presented here.

8) In section 5.5 it is mentioned that the analysis presented will focus on the increase above background in order to remove the bias in the background CO and PPM. This should be expanded to all the analysis, including surface stations.

9) There is no agreement between the comparison with surface and satellite observations and the reasons presented here to explain the differences diverge. This analysis lacks clarity.

Other comments:

Throughout the whole document the word enhancement is used in the context of increase. Enhancement usually relates to improvements in quality not in increases, of for example fire contributions.

L178-179: It is mentioned that "a matrix of correspondence between the MODIS IGBP

and the CLC vegetation types is provided." Does this follow any of the correspondence matrix available in the literature? For example, Pineda, N., Jorba, O., Jorge, J., and Baldasano, J.: Using NOAA AVHRR and SPOT VGT data to estimate surface parameters: application to a mesoscale meteorological model, Int. J. Remote. Sens., 25, 129–143, 2004

L188-L189: It is mentioned that "This option has been developed after strong discrepancies in the daily variability in fire activity" ... Could you please reference a study providing details on this, or summarise the outcomes? This would allow a better understanding of the improvements given the discrepancies found in this study. L266: The analysis focuses on regional transport (around 100 km away from fires) and not long-range (> 1000 km)

L277-279: The reference to the results for southern France are solely mentioned here and do not add relevant information to this study if this is not used as a comparison between the two case studies further along.

L300: "Considering all experiments, it is equal to 23% on average during the main fire event". What is equal to 23%? Is it the average coefficient? What information does this provide? The paragraphs around this statement should be re-written in a clearer way.

L368-369 the way this is written implies that fire emissions are also assumed to be more intense during the day and an emission scaling is applied. Is this also applied to fire emissions? Please re-write this paragraph in a clearer way.

L396: The paragraph starting in this line is very confusing to read and mixes two distinct ideas, please reformulate this.

L465-468 Were there any significant dust events prior or during the period of study that corroborate this statement?

Paragraph stating on line 470 suggests that having a better representation of the injection heights is important for the transport and vertical distribution of the emissions. Would the authors consider adding a comparison with LIDAR data (e.g. from CALIPSO)?

L502-509: Does this means that due to the lack of sensitivity of the satellite to these levels, this satellite data is not fit for purpose?

---

## Referee Comment (RC2) · Anonymous Referee #2 · 2 Dec 2019

This study presents an updated version of the APIFLAME model to estimate biomass burning emissions. The new biomass burning (BB) emissions dataset was evaluated by using the offline air quality model CHIMERE for summer 2016, focusing on the intensive wildfires in Portugal. The major update in APIFLAME is merging the burned area and fire radiative power datasets in order to simulate the temporal variability of the BB emissions and also incorporate the small fires. Various configurations of the APIFLAME model are presented here to show the range of the uncertainties in the BB

emission estimates.

Development of accurate and fine-scale BB inventories is crucial for assessing the impact of fires on air quality and climate. The methods used in the study are novel. However, the paper requires major improvement for publication in GMD.

First, some of the wording in the text has to be improved. Some sentences aren't clear.

Other major comments:

I suggest changing the title of the paper to state that this study isn't only evaluating the BB emissions, but also the CHIMERE model. I know in a number of studies BB emissions were evaluated by using the atmospheric models. However, this gives a false impression that different BB emissions can be evaluated accurately by plugging them into the air quality models. As the authors note, there are many uncertainties in the modeling of the plume injection height, tracer transport and mixing, and atmospheric chemistry in the air quality and atmospheric chemistry models. These uncertainties have profound effect on the performance of the atmospheric models. This point has to be made clear in the Abstract as well.

It has to be emphasized that the APIFLAME2.0 can be used for the retrospective studies, not for forecasting. There are significant challenges in forecasting BB emissions, especially on the regional scales. Sometime it's assumed that using the new satellite data the forecasting of BB emissions can be easily done. The satellite FRP data provide information about the state of a fire intensity, unless the satellite scans are obscured by dense smoke or cloudiness. It's still very hard to forecast the spatial and temporal variability of the BB emissions for next hours and days.

The burned area data is the primary source of the information to estimate the BB emissions in APIFLAME. I suggest adding a short description of the burned area dataset and associated uncertainties.

In section 4.1 it's stated that high FRP points correspond to larger burned areas. While

this assumption is true in general, sometimes the MODIS instruments report very high FRPs (∼1000MW) for small agriculture fires, for example when the overpass time coincides with the intensive flaming phase.

One of the key uncertainties in modeling of the BB emissions is the estimating flaming and smoldering emissions. This topic isn't discussed in the paper. What emission factors (EFs) did the authors use to estimate the emissions of CO and other chemical species? For smoldering or flaming phases?

When the plume rise parameterization is used, how the emissions are partitioned in CHIMERE? What vertical distribution was used for the BB emissions in the model with Sofiev et al. plume rise parameterization?

L.405: The studies show that as the semi-volatile POA species emitted by fires evaporate partially, more SOA forms downwind of fires. Overall these two processes compensate each other. How these processes are parameterized in CHIMERE?

How the AOD is calculated from the CHIMERE model output? In the literature the reported aerosol extinction coefficients for smoke vary from one study to another. Was any hygroscopicity assumed for the modeled BB aerosols?

Some text and figures can be moved to SI, e.g. Figure 3, Tables A1-2.

Using the MISR data to constrain the injection heights in the model is a reasonable approach. However, the MISR typically misses the most intensive stages of the fire evolution (occurring during the afternoon hours). This will lead to underestimate of the fire injection heights overall.

Minor comments:

L. 540: Move "The ability. . ." to the Introduction.

---

## Author Comment (AC1) · 8 Mar 2020

**Reply to Referee Comment 1**

by S. Turquety et al.

The co-authors thank the referee for carefully reading the manuscript and for the helpful comments. We have addressed all of the comments (copied in italic letters below to facilitate reading). Corrections to the text are reported in blue. An important point to keep in mind is that this article is about the development of a biomass burning emissions model. Here, the WRF and CHIMERE models are just hosts to test these fluxes and it could be any other model. In his/her review, the referee has remarks for both modelling systems: (i) the fires emissions, (ii) the chemistry-transport model. If remarks about the CTM are correct and interesting, it is not possible and not the focus of the paper to make all sensitivity tests and all suggested calculations. In addition, and knowing all available CTMs, a lot of suggestions would probably have different answers depending on the CTM. In conclusion, we added a lot of new tests, calculations and discussion and we focused it on the main goal of this paper: the biomass burning emissions.

**Major comments**

*1) The way the manuscript is written, and structure is very confusing, making it hard to read through. I would recommend this to be reviewed by an English native speaker and I would advise restructuring the text in some of the sections in order to improve readability and conclusions.*

Many parts of the text have been rewritten, as described in our response to the different comments. In addition, we have modified the structure of the text, more specifically for the analysis of the case study which raised a lot of questions. Section 5 now follows the following outline:

5.1) APIFLAME biomass burning emissions

5.2) Observations of atmospheric concentrations

5.3) CHIMERE-WRF regional CTM

5.4) Sensitivity of the simulated concentrations to the configuration of APIFLAME

5.5) Evaluation against observations

(a) Surface observations

(b) Satellite observations of aerosols (MODIS, CALIOP)

(c) Satellite observations of CO (IASI)

Several figures have also been moved to supplemental material as suggested in the second referee comment (RC2), which also makes the text easier to read.

30 *2) Section 4 shows the methodology for merging burned area with radiative fire power (RDP) data in order to increase the temporal variability which is crucial for estimating emissions for this study. It is shown how the RDP of small fires contributes to an increase of the burnt areas estimates. It would add value to this analysis to include an assessment on how this could impact fire emissions (e.g. using datasets such as GFAS).*

We have included an estimate of the resulting increase in emissions, as calculated with APIFLAME, for the example of CO, 35 at the end of Section 4.1. A small correction in the calculation resulted in slightly different values of burned area contributions, the corresponding numbers have been corrected (L.224–227). This does not impact Figure 3 (which was moved to supplementary material figure S1 as suggested in RC2).

The modulation using the FRP value used in this study is a strong approximation that should be used with caution. It results 40 in large potential increase in calculated burned area over most regions: ∼46% in boreal and in temperate North America, 37% in Equatorial Asia, 16–22% in boreal Asia, South-East Asia, Europe, Central and southern hemisphere South America. In Africa and Australia, calculated contributions are lower (∼5%) due to the low FRP of active fires non collocated with a MCD64 detection. (Randerson et al., 2012) estimate an increase of burned area from small fires ranging from 7% in Australia (5% in this study) to 157% in Equatorial Asia (37% in this study) for the 2001-2010 time period based on the MODIS collection 5 45 fire products. They generally found much higher contributions, for example in temperate North America (75%) and Europe (112%) or boreal Asia (62%). However, the collection 6 product (used here) has been shown to detect more fires (26% increase in global burned area over the period 2001–2016) with better coincidence with active fire products (68% within two days) (Giglio et al., 2018). For example, mean annual burned area for the period 2002–2016 in Europe is 71% higher in collection 6 than in collection 5.

50 Since the methodology used here relies on the burned area (Equation 2), the increase of burned area by small fires directly affects emissions. However, the relationship is not linear as it depends on the vegetation type attributed. For CO for example, emissions are increasing by ∼60% in North America, ∼20% in Central and South America, ∼25% in South-East Asia, ∼23% in Europe, ∼5% in Africa and Australia.

55 *3) It is mentioned in L288-289 that for MIDIS IGBP "limit uncertainties in APIFLAME, fires detected in areas of chaparral or mediterranean vegetation types and classified in shrubland but also savanna are attributed to chaparral". Further along in L295- 298 the authors state that for CLC forcing the biome of temperate forest allows a better agreement between the results of the experiments. Why not use a consistent approach and use the different datasets to understand the sensitivity to uncertainties regarding the land surface categorization?*

60 *6) In the paragraph between lines 440 and 457 it is mentioned that regional fire emission reports from Alves et al. (2011) are larger than the ones used in this study. Considering that there are specific emission factors in the literature for the study region*

*why was not do an assessment of the sensitivity of these fire emissions compared to the values proposed by other authors and the ones used in this work? Would the conclusions of this study be affected by the choice of the emission factors and land cover database? These are topics that are mentioned in this paper but are not explored, which would add value to this paper.*

65

We appreciate the helpful suggestions. We have added two sensitivity tests regarding: 1) the vegetation database, using either CLC or MODIS (impact of landcover database), 2) the emissions factors using either tabulated average values of the results from Alves et al. (2011).

We have also significantly revised the structure in order to more clearly describe the experiments conducted as well as the 70 resulting differences. Results from the sensitivity simulations are now presented in a dedicated Section 5.4 "Sensitivity of the simulated fire plume to the configuration of APIFLAME" and summarized in a table, together with differences in emissions.

Maps showing the relative difference in the contribution from fires for each factor tested have been added in the text for CO (Figures 1). Figure 12, which shows the coefficient of variation calculated based on all the sensitivity simulations, has been updated to account for the new sensitivity tests.

75 The results are discussed in a dedicated subsection (5.4):

The average sensitivity of the surface and total CO and PM10 to the factors influencing the biomass burning emissions tested in this study are reported in Table 3. The average differences are calculate for the fire plume, over points with contribution from fires >10% The maps for CO are presented in Figure 11.

80 As for the emissions, the largest impact is associated with the small fires (increased burned area) and the higher emission factors. For small fires, the relative difference is equal to 41%–48% on average in the denser part of the plume (relative contribution >10%). Using increased emission factors, the CO concentrations are more than doubled (×2.5 at the surface, ×2.1 for the column). The impact on PM10 is lower (lower increase in EF) but still almost as high as the contribution from small fires. Both effects are particularly marked for surface contributions. The influence of the different vegetation types is more nuanced. 85 Using the CLC database results in an increase of CO in Portugal but not in other regions. The effect is lower than that of small fires and emission factors but still very significant, on average 10–30%. These influences are in line with the sensitivity of emissions discussed in Section 5.1.

The sensitivity to injection heights is also presented, using either the MISR profile (resulting in ∼25% of emissions injected above the PBL) or the default scheme in CHIMERE (for this case, all emissions in the PBL). Having a fraction injected above 90 the PBL mechanically decreases surface concentrations (by ∼25%). The total column tends to increase to the North of the fire region, and decrease to the South. This shows that injection height has a significant impact on transport pathways. Northward transport (towards the Bay of Biscay) will tend to be in the free troposphere, while southward transport remains at low altitude.

*4) In section 5.4 it is said that Secondary Organic Aerosols (SOA) have a very low fraction and can be discarded as a 95 justification for not accounting for these. However, this is later presented as a limitation of this study affecting the bias of the modelling approach.*

**Table 1.** Relative impact (%) of factors tested in the calculation of the emissions with APIFLAME on the total emissions of CO and organic carbon, and on the resulting concentrations. Differences of simulated concentrations ($\Delta CO$ and $\Delta PM10$) are averaged over points with relative impact from fires $>10\%$.

| Factor | Simulations compared | $\Delta E_{CO}$ | $\Delta E_{OCAR}$ | $\overline{\Delta CO}$ | | $\overline{\Delta PM10}$ | |
|---|---|---|---|---|---|---|---|
| | | | | surface | total | surface | total |
| Merge BA-FRP | BA-FRP-MODIS − BA-MODIS | +10 | +8 | - | - | - | - |
| Small fires | BA-sf-MODIS − BA-MODIS | +33 | +36 | +43 | +48 | +41 | +47 |
| Vegetation | BA-sf-MODIS − BA-sf-CLC | −17 | +0.3 | −30 | −17 | −11 | −29 |
| Fuel consumption | BA-sf-MODIS − BA-sf-MODIS-lit | +2 | −2 | - | - | - | - |
| Emission factor | BA-sf-MODIS-EF − BA-sf-MODIS | +126 | +50 | +152 | +118 | +40 | +14 |
| Injection height | BA-sf-MODIS MISR − BA-sf-MODIS | - | - | −25 | +11 | −22 | +32 |

SOA production is included in the CHIMERE CTM. However, in the version of the model used here, the contribution in the fire plume is small, as shown in Figure 11. The implementation of new pathways for SOA formation, from semi-volatile VOC and VOC of intermediate volatility (I-SVOC), shows significantly higher values ($\sim$30%), as detailed in Majdi et al. (2019). In this study, we have chosen to use the distributed version of CHIMERE, which does not include the formation of SOA from these species.

In Section 5.4, we do not suggest that SOA may be discarded but that it is most likely underestimated. In the study of Majdi et al. (2019), we show that SOA formation from I-SVOC could explain the missing mass of aerosols in the inventories of emission factors (difference between the sum of emission factors for the identified aerosols and for PM2.5). In APIFLAME, this missing mass is attributed to a surrogate species ($EF_{other\ fine\ PPM} = EF_{PM2.5} - \sum_{i=aerosols} EF_i$). Therefore, if SOA formation from I/S-VOC is included in a CTM, emissions of this PPM$_{other}$ species should not be included to avoid possible double counting.

In order to clarify this, we have replaced the paragraph:

"As explained in Section 3.4, the contribution from other PPM, here assumed to be inert fine particles, could correspond to SOA produced by intermediate and semi-volatile organic compounds that are not accounted for in the current version of CHIMERE (Majdi et al., 2019). Indeed, SOA here contribute to a very low fraction and are most likely underestimated. Emissions from this surrogate PPM can be discarded if sufficient SOA is formed."

by:

"As explained in Section 3.4, the surrogate species other PPM (inert fine particles) is introduced to account for the missing mass of aerosols in the inventory (difference between the emission factor of PM2.5 and the sum of emission factors for the identified aerosols). Majdi et al. (2019) have shown that its contribution to atmospheric concentrations of aerosols is of the

[Figure]

**Figure 1.** *Relative impact of biomass burning emissions on surface concentrations and total columns of CO (simulation $BA$), and sensitivity to each factor tested in the sensitivity simulations. Each row corresponds to the impact of one specific parameter, calculated using the difference between two sensitivity simulations as described in Table 1. Left maps correspond to the impact at the surface, right maps to the total columns. Note the change of scale for the impact of EF on the surface concentration of CO.*

same order of magnitude as the SOA produced by organic compounds of intermediate volatility and semi-volatile organic compounds (I-SVOC). In the simulations presented here, the contribution of I-SVOC to SOA formation is not included and the fraction of SOA in the biomass burning plume is very low, most likely underestimated. For model versions including SOA formation from I-SVOC, the contribution from the surrogate PPM species should not be taken into account in order to avoid double counting. "

5) This study uses climatologies as boundary conditions for the chemistry and aerosol species. This could have a significant impact on the background levels of these species and contribute to the bias presented away from fire regions. Considering this,

*what would be the impact of using real time boundary conditions in the conclusions here presented? This is only pointed as a limitation of this study.*

The main focus of the manuscript is on the methodology and on the description of the APIFLAME model. The case study is used to show that the model gives realistic and useful results, and also highlight the sensitivity of the results to the configurations used in APIFLAME. For this purpose, we have chosen to use a classic configuration of the CHIMERE model, with a climatology as boundary conditions, and focus on the analysis of the contribution from the fire event within the domain.

*7) It is stated that if not explained by excessive emission factors, the overestimate of CO concentrations in simulations can be explain by other factors such as the temporal variability of emissions, problems in representing the Planetary Boundary Layer or injection of fire burning plumes. Do these factors have a greater influence than the uncertainty in the plant functional types? This is especially relevant since the plant functional type determines the emission factor and the fuel load available for emissions. Having an analysis of these contribution would significantly increase the value of the results presented here.*

The overestimate of CO concentrations may be due to several factors affecting the simulation. It may be due to the emissions model itself with the emissions factors, but also the horizontal resolution, for example, or all other APIFLAME input parameters. It may also be due to the host models, WRF and CHIMERE, by the way of transport, mixing, chemistry, deposition. We understand the question of the referee to better identify all under- or over-estimation, but it is not possible as it. Uncertainties are present in all processes in a model and they can increase or decrease an error compared to measurements. Often, error compensations are also present, increasing the difficulty of the analysis. However, we tried to add significant sensitivity tests in the paper in order to have a clearer idea of why CO concentrations are overestimated. These tests were also performed to answer the other questions of this referee. And a synthetic sentence was added in the conclusions.

*8) In section 5.5 it is mentioned that the analysis presented will focus on the increase above background in order to remove the bias in the background CO and PPM. This should be expanded to all the analysis, including surface stations.*

We have added the following paragraph to the introduction of Section 5: "The analysis of the simulations is focused on the increase due to the wildfires event. A sensitivity study allows an analysis of the influence of several key factors of the calculation of fires emissions: burned area, vegetation type, emission factors and emission injection profiles. The simulations are then evaluated by comparison to surface and satellite observations. "

*9) There is no agreement between the comparison with surface and satellite observations and the reasons presented here to explain the differences diverge. This analysis lacks clarity.*

We have rewritten these sections to make them easier to read. The comparison to surface observations is now separate from the discussion of the variability of the different simulations, and the evaluation against satellite observations of aerosols and CO are presented in different sections (cf. response to the first comment).

**Other comments**

*Throughout the whole document the word enhancement is used in the context of in- crease. Enhancement usually relates to improvements in quality not in increases, of for example fire contributions.*

This has been corrected throughout the text. L.10: " Corresponding enhancements of aerosols and carbon monoxide (CO) simulated with the regional chemistry transport model CHIMERE" replaced by "The corresponding increase in aerosol and carbon monoxide (CO) concentrations simulated with the regional chemistry transport model CHIMERE"

L.398: "The strong enhancement..." replaced by "The strong increase of the fire contribution..."

L.428: "Both observations and simulations show two main enhancements..." replaced by "Both observations and simulations show two main increases during the fire event."

L.433: "some enhancements associated with filtered peaks..." replaced by "some values associated with filtered peaks..."

L.489: " Observed enhancements above fires..." replaced by "Observed increases above fires...".

*L178-179: It is mentioned that "a matrix of correspondence between the MODIS IGBP and the CLC vegetation types is provided." Does this follow any of the correspondence matrix available in the literature? For example, Pineda, N., Jorba, O., Jorge, J., and Baldasano, J.: Using NOAA AVHRR and SPOT VGT data to estimate surface parameters: application to a mesoscale meteorological model, Int. J. Remote. Sens., 25, 129-143, 2004*

There is generally a direct correspondance between the two land cover datasets. Otherwise, the correspondance was chosen using their description. For IGBP vegetation type "woody savanna", described as 30-60% tree cover, the CLC correspondance chosen is 30% mixed forest and 70% natural grassland. For IGBP "cropland" type, described as at least 60% of cultivated cropland, the corresponding CLC is 40% arable land and 60% permanent crop. For IGBP "Cropland/Natural Vegetation Mosaics" corresponding to mosaics of small-scale cultivation (40-60%) with natural tree, shrub, or herbaceous vegetation, the corresponding CLC types chosen are 50% "mixed cropland and other vegetation" and 50% "mixed cropland and forest".

The following sentence was added to the text:

"When there is no direct correspondance between the two land cover datasets, their description was used. For example, for IGBP vegetation type "woody savanna", described as 30-60% tree cover, the CLC correspondance chosen is 30% mixed forest and 70% natural grassland.

*L188-L189: It is mentioned that "This option has been developed after strong discrepancies in the daily variability in fire activity" ... Could you please reference a study providing details on this, or summarise the outcomes? This would allow a better understanding of the improvements given the discrepancies found in this study.*

We came across such discrepancies for several fire events that we have looked at. This issue is introduced in Section 2, L.82, "burned area and FRP are not always detected at the same time or location in the MODIS datasets. If the burned area dataset is chosen, there may be grid cells with non zero burned area (non zero emissions) but zero FRP (or the other way around).

Merging both datasets may be an interesting option for some applications, for example to improve temporal variability or in order to avoid missing small fires that may not have a detectable signature on both products."

We can also see this in Figure 7 (section 5.2) which shows a different daily variability during the 2016 event in Portugal between the MCD64 "BA" and merge "BA-FRP" configurations.

We have replaced the sentence L188-189 with:

Differences in the location of burned areas and active fires detected were found for different events in different regions.

*L266: The analysis focuses on regional transport (around 100 km away from fires) and not long-range (> 1000 km).*
This has been corrected.

*L277-279: The reference to the results for southern France are solely mentioned here and do not add relevant information to this study if this is not used as a comparison between the two case studies further along.*
This sentence has been removed.

*L300: "Considering all experiments, it is equal to 23% on average during the main fire event". What is equal to 23%? Is it the average coefficient? What information does this provide? The paragraphs around this statement should be re-written in a clearer way.*
This paragraph has been rewritten to clarify this:

The dispersion of the regional daily total emissions is quantified as the average coefficient of variation (CV = standard deviation / mean value). This provides an information on the uncertainty on the emissions. Considering all experiments, the CV on the total daily emissions of CO, averaged over the duration of the fire event is equal to 40%. Without the experiments including small fires, it is 15%. Without the experiments with emission factors from (Alves et al., 2011), it is 20%. At 10 km resolution over the full domain (hence without summing all emissions within the region), the average CV is around 80% on daily emissions, 60% without experiments including small fires, 76% without experiments on emission factors.

*L368-369 the way this is written implies that fire emissions are also assumed to be more intense during the day and an emission scaling is applied. Is this also applied to fire emissions? Please re-write this paragraph in a clearer way.*

I am not sure to understand this comment correctly. The emissions are not scaled, the total daily is redistributed throughout the day. This sentence has been rewritten.
" So that 70%..." has been replaced by:

The total daily emissions are thus redistributed over the day (the total remaining unchanged), assuming that 70% of the total will be emitted during the day between 8am and 8pm local time, and the remaining 30% at night. "

*L396: The paragraph starting in this line is very confusing to read and mixes two distinct ideas, please reformulate this.*

The comparison with satellite observations has been rewritten to improve clarity: *Cf.* response to major comment (9).

*L465-468 Were there any significant dust events prior or during the period of study that corroborate this statement?*

During summer and in the south of Europe, there is always long-range transport of mineral dust from Africa. Regional CTMs may have difficulties to reproduce these events, and this is the case of CHIMERE as every models. Many publications exist to model these events, including with CHIMERE. This is not the focus of this paper to discuss this topic. But we agree that mineral dust may influence the AOD measurements and then be mixed with biomass burning products. To have an independent information about the presence of dust or not, we used the Giovanni NASA database (https://giovanni.gsfc.nasa.gov/giovanni/) to extract dust simulations by the MERRA-2 model (including assimilation). Using mineral dust surface concentrations, total column mass and Angstrom exponent, we found significant dust transport over the studied area at the beginning of the period. It is also clear that the background underestimation diagnosed between MODIS and CHIMERE is not only due to a mineral dust plume missing in the model. The peaks on the 19/07 and around 24/08 also correspond to dust transport from North Africa.

The sentence:

"Compared to observations, the simulated background AOD and total CO levels are too low, more particularly over Spain for CO, over the northern part and the mediterranean area for AOT. This could be due to underestimated local emissions but also to the use of a climatology as boundary conditions, which does not allow inflow due to long-range transport (for example from dust outbreaks from North Africa for AOD)."

has been replaced by:

" An underestimate of AOD is diagnosed with the model over the Atlantic and the Mediterranean as well as over Spain and France. This underestimate may be due to a lack of emissions (anthropogenic, biogenic, fires or mineral dust) or to the way AOD is calculated in the model (using the Fast-JX online model in CHIMERE). A classical candidate of this kind of underestimation in regional models and in summertime is a missing mineral dust plume coming from Africa. Gama et al. (2020) show that desert dust contribute significantly to PM concentrations in Portugal and are mixed with the fire contribution in 2016. The analysis of mineral dust concentrations and Angstrom exponent using global databases displayed with Giovanni (https://giovanni.gsfc.nasa.gov/giovanni/, Figure S2) confirms that there is a long-range transport of mineral dust above the Atlantic from Africa to Europe at the beginning of the fire event (06-09/08). It increases AOD above the Atlantic, Northern Spain and the Bay of Biscay on 08/08 and Southern Portugal on 09/08, and will be mixed with the biomass burning contribution. However, there is no transport towards the Mediterranean. Since this is a background bias, homogeneous over land and ocean, and not a problem of plume with high values, this modelling problem has to be investigated more generally with the CHIMERE model but is not due to the biomass burning inventory presented in this study."

[Figure]

**Figure 2.** *Mineral dust total mass density from the global databases displayed with Giovanni (https://giovanni.gsfc.nasa.gov/giovanni/; MERRA-2 model simulations including data assimilation).*

Paragraph stating on line 470 suggests that having a better representation of the injection heights is important for the transport and vertical distribution of the emissions. Would the authors consider adding a comparison with LIDAR data (e.g. from

*CALIPSO)?*

265

Following the referee's suggestion, we have decided to add comparisons to CALIOP VFM product. We have added at the end of Section 5.2 for the constrain on injection height:

The vertical distribution of aerosols is studied using observations from the Cloud-Aerosol Lidar with Orthogonal Polarization (CALIOP), on-board the Cloud-Aerosol Lidar Pathfinder Satellite Observation (CALIPSO) satellite (Winker et al., 2009).

270 Here, the Vertical Feature Mask (VFM) Level 2 product V4-20 (Winker, 2018) is used in order to identify the altitude of the aerosol layers and their dominant type (Kim et al., 2018). The classification includes two subtypes for the identification of aerosols from biomass burning: polluted continental/smoke, corresponding to non-depolarizing aerosols within the PBL (mixing both polluted continental aerosols and biomass burning aerosols which have close optical properties) and elevated smoke (layers with tops higher than 2.5km, which may include non-smoke pollution lofted above the PBL).

275

In Section 5.3:

"LIDAR observations also allow the analysis of aerosol plume height and have been used in several studies to analyse injection height (e.g. Labonne et al., 2007). However for this case study, there was no CALIPSO overpass above the fire region so that there could be no constrain on the profile at emission. "

280

We have added the following analysis to Section 5.5.2- Satellite observations of aerosols (MODIS, CALIOP):

The altitude of the aerosol plumes is analyzed using the aerosol layer classification from CALIOP (VFM product). Four CALIPSO overpasses are available during the studied time period:

- 08/08, 2:44UTC and 13:24UTC, both located above the Atlantic and capturing the outflow from the fires in North
285     Portugal;

- 10/8 13:11UTC, the only one above the continent but to the east of the fire region. It captures the recirculation above south-western Spain;

- 15/8 2:50UTC, located above the Atlantic, it captures the outflow to the North of the domain, at the edge of the domain so that part of the plume is lost in the simulations.

290 The observed VFM on 08/08 and the simulated PM10 concentrations along the CALIPSO track are shown in Figure 3 for the nighttime overpass, and Figure 4 for the daytime overpass. A large contribution from dust is observed in the free troposphere (2-7km) at latitudes >42°N, confirming that AOD at the beginning of the fire event corresponds to a mix of smoke and dust, the latter being underestimated in the simulations. The large contribution from fires simulated around 42N up to ∼4km is attributed to clean marine aerosols in the VFM (due to the color ratio). The 1064nm backscatter is high and may actually correspond to
295 the fresh smoke outflow. It is observed with similar structure but at slightly lower altitude by CALIOP. The smoke contribution in the southern part of the plume is observed at higher altitude with CALIOP (up to 3km) compared to the model (main contribution below 2km). For the daytime overpass, the observed plume is mainly attributed to dust, and probably corresponds to

mixed dust and smoke contributions. The model simulates a large contribution from fires around 3-4km, that is consistent with the observations, only if the MISR profile is used. On the 10/08, the tropospheric smoke aerosols are observed up to 2.5km around 38N, and up to 3.5km further North. For this case, the simulated layer height is in better agreement without the MISR profile, although slightly higher (up to 4km without MISR, 5km with MISR) around 39N. On the 15/08, CALIOP captures the plume at ∼1 to 4.5km. Using the MISR profile in the simulations increases this transport pathway (Figure 1). However, the plume is simulated too high using the MISR profile (1 to 6km), but too low without MISR (∼3.5km). Since this case These comparisons highlight the difficulty to simulate the fire plume height but also that using an averaged profile from MISR is a good option if no other observation is available.

[Figure]

**Figure 3.** *Vertical Feature Mask from CALIOP and corresponding CHIMERE simulations (BA-sf MISR) of total PM10 and PM10 from wildfires (08/08, nighttime overpass). The CALIPSO overpass track is plotted on a map of simulated total PM10 mass density from fires at the corresponding time.*

*L502-509: Does this means that due to the lack of sensitivity of the satellite to these levels, this satellite data is not fit for purpose?* The large CO plumes observed by IASI to be transported from the fire region shows that IASI is very sensitive to this event. Comparisons between remote sensing retrievals and model simulation outputs are difficult to analyze due to the sensitivity of the observing system (observation & retrieval algorithm). However, because of the lack of other observations more directly comparable to model simulations, such as in situ observations, these observations prove very valuable. They also offer a regional view of the transport pathways. We have re-written the section on comparison between simulations and IASI observations to hopefully clarify it and avoid wrong interpretation: *Cf.* response to major comment (9).

[Figure]

**Figure 4.** *Same as 3 but for the daytime overpass.*

We have reorganized the conclusions in order to better present the complementary between the surface and satellite observations:

The apparent conflict in the conclusions of the comparisons to surface observations (overestimated peaks) or satellite observations (tendency to underestimate vertically integrated values) has been found in several studies (e.g. Majdi et al., 2019; Carter et al., 2020) and may have several origins. First, the representativity of the surface data is low and transport errors will thus have very strong impact on comparisons. Secondly, transport may be simulated too low, or may be too wide on the vertical (due to the vertical resolution and numerical diffusion). The use of MISR plume height observations clearly improves comparisons, as already obtained in other model experiments (Rea et al., 2016; Zhu et al., 2018), but the lack of horizontal and temporal coverage of the instrument does not allow enough variability to represent the strong influence of fire intensity on injection heights. Comparisons to aerosol vertical structures observed by CALIOP show that the plume is transported at too low altitude towards the South (and at about the right altitude closer to the fire region). Too low vertical dispersion mechanically results in too high surface concentrations. For AOD, a desert dust transport event most probably contributes significantly to the observed AOD, and is not simulated in this study (the domain does not include North Africa).

For CO, the underestimate of the transported plumes is at least partly due to the altitude of the transport, which is very critical for comparisons to IASI observations. Indeed, IASI measurements are primarily sensitive to the free troposphere above oceans. Despite the limitations, due to the lack of in situ measurements over the region, and in general close to large fire events, the

good spatial and temporal coverage of these satellite observations provide very helpful information on the emissions spatial and temporal variability.

---

## Author Comment (AC2) · 8 Mar 2020

**Reply to Referee Comment 2**

by S. Turquety et al.

The co-authors thank the referee for carefully reading the manuscript and for the helpful comments. We have addressed all of the comments (copied in italic letters below to facilitate reading). Corrections to the text are reported in blue.

**Major comments**

*First, some of the wording in the text has to be improved. Some sentences aren't clear..*

Many parts of the text have been rewritten, as described in our response to RC1 and to the different comments. In addition, we have modified the structure of the text, more specifically for the analysis of the case study which raised a lot of questions. Section 5 now follows the following outline:

5.1) APIFLAME biomass burning emissions

10 5.2) Observations of atmospheric concentrations

5.3) CHIMERE-WRF regional CTM

5.4) Sensitivity of the simulated concentrations to the configuration of APIFLAME

5.5) Evaluation against observations

(a) Surface observations

15 (b) Satellite observations of aerosols (MODIS, CALIOP)

(c) Satellite observations of CO (IASI)

*I suggest changing the title of the paper to state that this study isn't only evaluating the BB emissions, but also the CHIMERE model. I know in a number of studies BB emissions were evaluated by using the atmospheric models. However, this gives a*
20 *false impression that different BB emissions can be evaluated accurately by plugging them into the air quality models. As the authors note, there are many uncertainties in the modeling of the plume injection height, tracer transport and mixing, and atmospheric chemistry in the air quality and atmospheric chemistry models. These uncertainties have profound effect on the performance of the atmospheric models. This point has to be made clear in the Abstract as well.*

We agree that the comparison of CTM simulation outputs with atmospheric observations of the biomass burning plume is
25 not a sufficient validation of the emissions. However, the illustration shows that the emissions calculated by APIFLAME allow a realistic simulation of the influence from biomass burning on regional air quality. The scope of the paper being primarily the description of the emissions model, we have removed the "evaluation against observation" from the title. To address the comments in RC1, we have added several sensitivity simulations and these now make up a large part of the results presented.

We have thus decided to change the title to:

APIFLAME v2.0 biomass burning emissions model: impact of refined input parameters on atmospheric concentration in Portugal in summer 2016

In the abstract, we have modified the last sentence:

"The overestimate compared to surface sites and underestimate compared to satellite observations point to uncertainties not only on emissions (total mass and daily variability) but also on their injection profile and on the modelling of the transport of these dense plumes."

to

The comparisons of the different CTM simulations to observations point to uncertainties not only on emissions (total mass and daily variability) but also on the simulation of their transport with the CTM and mixing with other sources. Considering the uncertainty on the emission's injection profile and on the modelling of the transport of these dense plumes, it is difficult to fully validate emissions through comparisons between model simulations and atmospheric observations.

*It has to be emphasized that the APIFLAME2.0 can be used for the retrospective studies, not for forecasting. There are significant challenges in forecasting BB emissions, especially on the regional scales. Sometime it's assumed that using the new satellite data the forecasting of BB emissions can be easily done. The satellite FRP data provide information about the state of a fire intensity, unless the satellite scans are obscured by dense smoke or cloudiness. It's still very hard to forecast the spatial and temporal variability of the BB emissions for next hours and days.*

As every model, it is possible to use APIFLAME in forecasting. We agree that the weak point in this case is the hypothesis of fires lifetime. Even if current observations of fire activity allow a real-time updating of past emissions, it has to be combined with other information to allow a realistic forecast. One possibility is the use of fire weather indexes, as in Di Giuseppe et al. (2017). We have added the following sentence in the conclusions:

APIFLAME may be used for near-real time applications (using MOD14 only). Forecasting the evolution of the emissions remains uncertain. However, the likelihood of a fire being controlled and extinguished increases when weather conditions are less favorable for its spread. A possibility is to modulate emissions with forecasts of fire weather indices computed from the forecasted meteorology, as suggested by (DiGiuseppe et al., 2017).

*The burned area data is the primary source of the information to estimate the BB emissions in APIFLAME. I suggest adding a short description of the burned area dataset and associated uncertainties.*

We agree that the description provided was probably not very clear and not well placed in the manuscript. We have moved and clarified paragraph:

The sentence in section 3.1: "During the processing of the fire observations, an estimated burned area is calculated for each burning pixel in the database as the pixel area actually covered by vegetation (Cf. section 3.2), as in the first version of API-FLAME (Turquety et al., 2014). "

is modified and moved to the beginning of Section 4.1:

"The fire observations described in Section 3.1 provide a date of burning at a resolution of 500m (MCD64 burned scars product) or 1km (MOD14 active fire product). The burned area is calculated for each burning pixel in the database as the pixel area actually covered by vegetation (Cf. section 3.2), as in the first version of APIFLAME (Turquety et al., 2014)."

*In section 4.1 it's stated that high FRP points correspond to larger burned areas. While this assumption is true in general, sometimes the MODIS instruments report very high FRPs (~1000MW) for small agriculture fires, for example when the overpass time coincides with the intensive flaming phase.*

We thank the referee for this comment. It is correct to say that it is not always the case, we just want to say it is the most commonly observed case. We agree that the approximations associated with this very simplified method are important and we have added a specific comment on this subject:

"A larger burned area is therefore associated with fires of greater radiative intensity. This may not be true if the satellite overpass coincides with the flame phase of the fire. Even a small fire can then have a high FPR. However, intense fires are expected to burn more fuel. The reported burned area should therefore be analyzed either as a larger area or as a larger burning fraction. This follows the same logic as the merging of burned scars and active fires. "

*One of the key uncertainties in modeling of the BB emissions is the estimating flaming and smoldering emissions. This topic isn't discussed in the paper. What emission factors (EFs) did the authors use to estimate the emissions of CO and other chemical species? For smoldering or flaming phases?*

We have used EF from tabulated reports available in the literature, as details in section 3.4. The numbers used do not separate flaming and smoldering phases. However, the reported combustion efficiency shows that flaming or smoldering may be favored depending on the vegetation. Since the inventory aims at being used in air quality models at resolutions of several kilometers, both flaming and smoldering phases should be mixed in one grid cell so that using average values seems relevant. Although it would be very interesting to be able to construct separate tables of EF for the flaming and the smoldering phases, it would then be difficult to have a systematic estimate of the fraction of flaming .vs. smoldering for each location and time in the region considered. This would require to develop methods using models and data at higher horizontal and temporal resolution.

We have added the following paragraph to Section 3.4 on emission factors:

"Although emission factors strongly depend on the phase of combustion (flaming favoring $CO_2$, $NO_X$ and $SO_2$, smoldering favoring CO, $CH_4$, $NH_3$, NMVOC and organic aerosols for example), the reported EF used in the model correspond to average numbers for different fire-type categories. Since the inventory aims at being used in air quality models at resolutions of several kilometers, both flaming and smoldering phases should be mixed in one grid cell so that using average values is relevant."

*When the plume rise parameterization is used, how the emissions are partitioned in CHIMERE? What vertical distribution was used for the BB emissions in the model with Sofiev et al. plume rise parameterization?*

A short description is added to section 5.3:

The injection profile is derived by assuming homogeneous mixing below this maximum height. Menut et al. (2018) have tested different shapes of injection profile in CHIMERE for the transport of biomass burning plumes in West Africa and show very low impact on the simulated concentrations.

*L.405: The studies show that as the semi-volatile POA species emitted by fires evaporate partially, more SOA forms downwind of fires. Overall these two processes compensate each other. How these processes are parameterized in CHIMERE?*
*How the AOD is calculated from the CHIMERE model output? In the literature the reported aerosol extinction coefficients for smoke vary from one study to another. Was any hygroscopicity assumed for the modeled BB aerosols?*

As it is not really the scope of the paper to fully describe the CHIMERE model, we added in the text a reference to the article of Couvidat et al. (2018). This article extensively describes the modeling of organic aerosols and the chemical mechanism leading to the formation of SOA.

The thermodynamic module ISORROPIA v2.1 (Fountoukis and Nenes, 2007) is used for inorganic aerosols and the module SOAP is used for organic aerosols (Couvidat and Sartelet, 2015).

and for the calculation of the optical properties:

"The optical properties of aerosols are calculated by the Fast-JX module version 7.0b (Bian and Prather, 2002). Fast-JX is used in CHIMERE for the online calculation of the photolysis rates."

*Some text and figures can be moved to SI, e.g. Figure 3, Tables A1-2.*
Done.

*Using the MISR data to constrain the injection heights in the model is a reasonable approach. However, the MISR typically misses the most intensive stages of the fire evolution (occurring during the afternoon hours). This will lead to underestimate of*

*the fire injection heights overall.*

We thank the referee for this comment and we have added a sentence about this limitation. We checked if there were any CALIOP observations above the fire region during the fire event (day time overpass being in the early afternoon) but the CALIPSO track was too far. CALIOP observations have been included in the revision for the evaluation of the height of the transported plume.

The following sentence has been included in section 5.2:
"Fires are usually at their highest intensity in the afternoon (corresponding to highest injection height). As the MISR observation is performed in the morning (Terra equator crossing time at 10:30LST), the fire plume height deduced will be quite conservative and will not correspond to a maximum. "

**Minor comments**

*L. 540: Move "The ability. . ." to the Introduction.*

Done.